# Hardware-Efficient Attention for Fast Decoding

**Ted Zadouri**[1,2]    **Hubert Strauss**[1]    **Tri Dao**[1,2]
[1]Princeton Language and Intelligence, Princeton University
[2]Together AI
`{tz6037,hs6702,td8762}@princeton.edu`

## Abstract

LLM decoding is bottlenecked for large batches and long contexts by loading the key-value (KV) cache from high-bandwidth memory, which inflates per-token latency, while the sequential nature of decoding limits parallelism. We analyze the interplay among arithmetic intensity, parallelization, and model quality and question whether current architectures fully exploit modern hardware. This work redesigns attention to perform more computation per byte loaded from memory to maximize hardware efficiency without trading off parallel scalability. We first propose Grouped-Tied Attention (GTA), a simple variant that combines and reuses key and value states, reducing memory transfers without compromising model quality. We then introduce Grouped Latent Attention (GLA), a parallel-friendly latent attention paired with low-level optimizations for fast decoding while maintaining high model quality. Experiments show that GTA matches Grouped-Query Attention (GQA) quality while using roughly half the KV cache and that GLA matches Multi-head Latent Attention (MLA) and is easier to shard. Our optimized GLA kernel is up to $2\times$ faster than FlashMLA, for example, in a speculative decoding setting when the query length exceeds one. Furthermore, by fetching a smaller KV cache per device, GLA reduces end-to-end latency and increases throughput in online serving benchmarks by up to $2\times$.

## 1 Introduction

In light of test-time compute (OpenAI, 2024), inference efficiency now drives progress in AI, demanding a greater emphasis on inference-aware architectures. The sequential nature of token-by-token decoding limits opportunities for parallelization. During decoding, Multi-Head Attention (MHA) (Vaswani et al., 2017) caches the key-value (KV) states of all prior tokens. These cached states scale linearly with batch size and sequence length and quickly exhaust high-bandwidth memory (HBM). Moreover, fetching this large KV cache from off-chip memory dominates execution time, significantly outweighing the relatively small computation performed by the matrix-vector workload at each decoding step. Memory fetches increase latency as the KV cache grows, resulting in prolonged cycles of low GPU utilization (He & Zhai, 2024). This bottleneck hinders a wide range of use cases: (i) latency-sensitive interactive applications; (ii) large batch LLM agents for multi-step reasoning (Yu et al., 2024b); (iii) test-time compute scaling (OpenAI, 2024; Snell et al., 2024); (iv) high-throughput batch inference; and (v) long-context video modeling (Wu et al., 2024). Collectively, these issues highlight the critical need for a hardware-efficient redesign of attention. An ideal attention mechanism should (1) achieve high model quality, (2) scale efficiently across multiple devices, and (3) utilize modern hardware effectively at inference time.

Inference-aware variants of attention accelerate decoding. Multi-Query Attention (MQA) (Shazeer, 2019) caches only a single KV head, significantly reducing memory usage. Grouped-Query Attention (GQA) (Ainslie et al., 2023) offers a compromise by sharing KV heads among smaller groups of query heads for better quality. Multi-head Latent Attention (MLA), recently introduced by DeepSeek (DeepSeek-AI, 2024; 2025), compresses the hidden state into a joint latent vector through low-rank factorized projections and caches the single latent head with large dimension, and then up-projects it before computing attention. These variants, as mentioned, reduce the KV cache, thereby easing MHA's memory-bound performance during decoding by lowering data movement (Ivanov et al., 2021; Ootomo &

Yokota, 2023; Gholami et al., 2024). Regrettably, the computational capabilities of modern accelerators have outpaced the growth of memory bandwidth (Ghose et al., 2018). Arithmetic intensity (Williams et al., 2009), the ratio of arithmetic operations to bytes of memory access (FLOPs per byte), is commonly used to analyze whether a workload is memory-bound or compute-bound. MQA increases arithmetic intensity by reusing a single KV head across all query heads, reducing the footprint of KV cache memory and therefore shortening its reload time from high-bandwidth memory while keeping FLOPs constant, but sacrificing quality and parallelism (Pope et al., 2022). GQA slightly increases the arithmetic intensity (proportionately to the group size) and scales efficiently during inference. However, with a moderate tensor-parallel degree, each GPU still stores a sizable KV cache, and model quality diminishes for large group sizes. Alternatively, DeepSeek absorbs its low-rank MLA projection matrices during decoding and directly uses the single latent head to compute attention, effectively doubling the arithmetic intensity relative to MQA. However, MLA inherently replicates the latent across all devices, limiting parallel inference.

In this work, we redesign hardware-efficient attention variants through the lens of arithmetic intensity, focusing on scaling during the decoding stage while preserving model quality. We define the group size $g_q$ as the number of query heads per distinct KV head; this group size largely determines the arithmetic intensity. Raising $g_q$ boosts the arithmetic intensity and proportionally shrinks the KV cache. However, past a threshold, further increases in $g_q$ begin to trade off higher operations per byte of memory access, effectively GPU utilization, against parallelization, forcing duplication of projection weights and KV cache across devices, diminishing parallel scalability. Guided by these design principles, we propose two attention variants that combine high arithmetic intensity with efficient scaling across devices, complemented by low-level optimizations. To translate these design choices into practice, our kernels overlap compute with memory through asynchronous software pipelining and warp specialization, employ a cooperative offset calculator for paged KV, and in doing so, keep tensor cores fully loaded, pushing the kernels from being memory-bound towards compute-bound.

- We initially explore Grouped-Tied Attention (GTA), which ties the key and value representations into one shared state that is used by small groups of query heads. GTA can reduce the KV cache size and improve the arithmetic intensity by up to a factor of 2 relative to its GQA counterpart with the same group size while preserving quality and parallelism.

- We propose Grouped Latent Attention (GLA), a parallel-friendly extension of latent attention that benefits from low-level optimizations. GLA achieves a similar quality to MLA and can be up to $2\times$ faster; for example, in speculative decoding, when the query sequence length is two or greater. In online serving benchmarks, GLA reduces end-to-end latency and increases token throughput by up to $2\times$.

- We demonstrate the efficacy of these variants in moderate-scale language modeling experiments trained on FineWeb-Edu. In an XL model (1.47B), GTA achieves a perplexity of 10.12 (versus 10.20 for GQA), and GLA reaches a 60.0% average downstream accuracy with 10.21 perplexity (vs. 59.1% and 10.25 for MLA). In a large model (876M), GTA achieves 11.2 perplexity and 57.6% average downstream accuracy (improved on GQA's 11.3 and 56.9%). For a medium model (433M), GLA yields an average downstream accuracy of 55.4%, slightly above MLA's accuracy of 54.9%.

- We combine their algorithmic design with a collection of low-level optimizations, leading to attention-decoding kernels 1.2-2$\times$ faster than FlashMLA [1] (Li, 2025). We release these optimized kernels under a permissive open-source license to benefit researchers and practitioners. The code is available at: `https://github.com/Dao-AILab/grouped-latent-attention`

## 2 Preliminaries

### 2.1 Inference-Aware Attention

MQA (Shazeer, 2019) reduces the KV cache size in MHA, thus accelerating decoding. It caches one KV head shared by all query heads. However, this aggressive sharing sacrifices model quality, and during distributed inference that partitions work at the head level, each device must replicate the single KV head to keep it accessible, thereby negating memory

---

[1] Benchmarks were performed using the FlashMLA kernel version dated 28 March 2025.

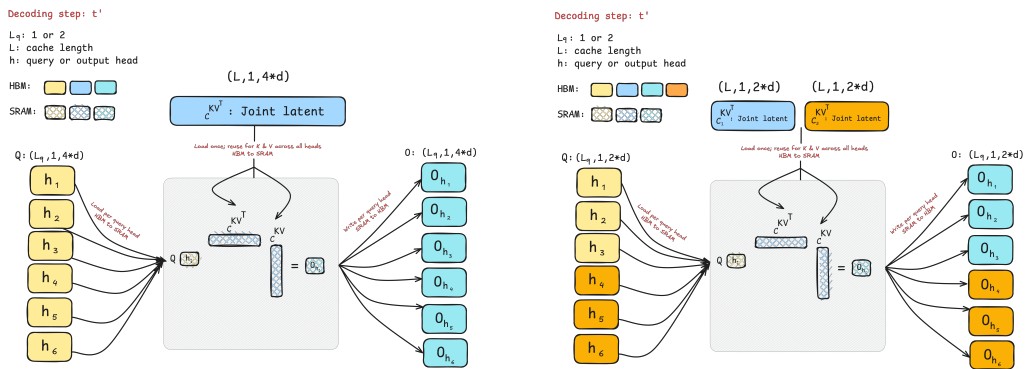

Figure 1: Memory-loading schematics during decoding of MLA **(Left)** and GLA-2 **(Right)** illustrate reduced data movement and higher arithmetic intensity, achieving more FLOPs per byte accessed and easing the memory-bound bottleneck. In MLA, single latent head $c^{KV}$ with $d_c = 4d_h$ is loaded once from HBM to SRAM and reused as $K$ and $V$ for every query head $\sigma(QK^\top)V$. In GLA-2, two latent heads, each with $d_c = 2d_h$, are likewise loaded once and reused as $K$ and $V$ for every query in their groups, eliminating or mitigating cache duplication when queries are sharded across devices.

savings (Pope et al., 2022). GQA (Ainslie et al., 2023) addresses these issues by grouping query heads to share a distinct KV head. This approach eliminates most duplication across GPUs at no extra memory cost in distributed inference. As a result, it reduces the KV cache size relative to the standard MHA while improving the model quality over the MQA. To preserve quality, it needs a moderate number of groups; as a result the per-GPU KV cache for long sequences or large batches can quickly exceed HBM capacity when using only a few devices, e.g., with a tensor parallelism degree 2. The KV cache per device only shrinks substantially when using many devices, e.g. an eight-way split.

Multi-head Latent Attention (MLA), initially introduced in DeepSeek V2 (DeepSeek-AI, 2024) and repopularized through its use in DeepSeek R1 (DeepSeek-AI et al., 2025) and FlashMLA (Li, 2025), compresses the hidden state of each token into a low-rank latent vector $c^{KV}$, caches only this vector, then projects it back to full-head keys and values to preserve distinct per-head features. For further reduction, its design only caches a single-head low-rank joint representation for both keys and values instead of separate ones. During decoding, the up-projection of the key $W^{UK}$ is absorbed in the query matrix $W^Q$, and the up-projection of the value $W^{UV}$ in the output matrix $W^O$. As a result, explicit keys or values never materialize; instead, each query attends directly to the latent $c^{KV}$. MLA keeps positional information outside the compression path by concatenating a small decoupled Rotary Position Encoding (RoPE) (Su et al., 2023) with the latent, allowing the weight-absorption trick. The resulting prefill attention is $\sigma(QK^T + Q_{rope}K_{rope}^T) \cdot V$.

## 2.2 Distributed Inference

Sequential decoding limits parallelization opportunities to only the head axis for the attention component. Tensor parallelism (TP) partitions attention layers across devices with frequent synchronization (e.g., all-gather) of activations. This overhead is mitigated by fast GPU interconnects such as NVLink (NVIDIA Corporation, 2024). TP avoids replicating the entire model per GPU, unlike Data Parallelism (DP) (Li et al., 2020), which becomes impractical for large models. It also circumvents idle GPUs during token-by-token decoding, a limitation seen in Pipeline Parallelism (PP) (Narayanan et al., 2021), which splits the model into sequential stages. Tensor parallelism is preferred during decoding because it distributes weights across GPUs, reducing the bottleneck of loading the KV cache when processing one token at a time (Su et al., 2025).

## 3 Methodology

We describe our perspective on designing hardware-efficient attention variants by focusing on the arithmetic intensity. We then demonstrate how to maximize the arithmetic intensity while efficiently parallelizing across devices by tying the key and value states and sharding the cached latent representation.

### 3.1 Arithmetic Intensity in Decoding: A Hardware-Efficient Perspective

During sequential decoding, the large GEMM workload (General Matrix Multiplication) used during training or prefill shifts to smaller batched GEMV (General Matrix-Vector Multiplication). Every loaded BF16 (2 bytes) element of the cached key matrix performs one MAC (2 FLOPs) with the single-token query element already in registers, yielding a 1:1 FLOP-to-byte ratio. This arithmetic intensity is far below the dense BF16 roofline of an Nvidia Hopper H100 SXM GPU (NVIDIA, 2022), $\sim$295 FLOPs per byte ($\frac{989\,\text{TFLOPs}}{3.35\,\text{TB/s}}$), leaving the tensor cores severely underutilized. In essence, the overall latency in distributed inference is limited by whichever is slower: the time to complete all FLOPs at the GPU's peak compute, the time to transfer the necessary data at the GPU's peak memory bandwidth, or the delay from inter-device communication at the available inter-connect bandwidth (Pope et al., 2022; Austin et al., 2025). In practice, with a modest TP degree, e.g., eight-way shard, repeated loading of the large KV cache dominates decoding latency (Recasens et al., 2025). The GPU utilization of standard MHA, where the arithmetic intensity is $\sim$1 as shown in Table 1, can drop to as low 7% during decoding (Recasens et al., 2025). In theory, it could accommodate around two and a half orders of magnitude more FLOPs without increasing latency.

| Attention Variant | GLA-2 | GLA | MLA | MQA | GQA | GTA | MHA | General Formulation |
|---|---|---|---|---|---|---|---|---|
| **Arithmetic Intensity** | $\frac{L}{1+\frac{L}{h_q}}$ | $\frac{L}{1+\frac{L}{2\cdot g_q}}$ | $\frac{L}{1+\frac{L}{2h_q}}$ | $\frac{Lh_q}{h_q+L}$ | $\frac{Lh_q}{h_q+\frac{h_q}{g_q}L}$ | $\frac{2Lh_q}{2h_q+\frac{h_q}{g_q}L}$ | $\frac{L}{1+L}$ | $\frac{2\cdot L}{2+\frac{m_{kv}}{g_q}L}$ |
| | $\approx h_q$ | $\approx 2g_q$ | $\approx 2h_q$ | $\approx h_q$ | $\approx g_q$ | $\approx 2g_q$ | $\approx 1$ | $\approx \frac{2g_q}{m_{kv}}$ |

Table 1: Let $L$ be the KV sequence length, $h_q$ the number of query heads, $h_{kv}$ the number of KV heads, and define the group size $g_q = \frac{h_q}{h_{kv}}$ (queries per KV head). The KV multiplicity is $m_{kv} \in \{1,2\}$, with $m_{kv}=1$ for shared KV states ($K=V$) and $m_{kv}=2$ for distinct KV states ($K \neq V$). We assume $L \gg h_q$.

### 3.2 Design Strategies for Maximizing Arithmetic Intensity

GQA is a general attention formulation that reuses one loaded KV head across each group of query heads. Since sharing the KV head does not reduce the number of operations (each query head still computes its attention scores) but does reduce memory reads, the arithmetic intensity increases proportionally to the group size $g_q$ or queries per distinct KV head (see Table 1). For example, in the case of MQA, which reuses a single KV head across all query heads, the arithmetic intensity is approximately the number of query heads, $h_q$.

DeepSeek introduced MLA with an inference-focused design that absorbs low-rank matrices that avoid materializing KV states, thereby achieving high arithmetic intensity through three factors: caching a single head latent, reusing the latent for both key and value, and employing a large number of query heads. Initially, MLA loads a single latent head and reuses it across all query heads, similar to MQA. The exact latent representation loaded into on-chip memory now serves both the key and the value states when computing attention, effectively doubling the arithmetic intensity compared to the most aggressive MQA designs (see Figure 1, left, for an illustration of the MLA memory loading schematic). Although the DeepSeek paper (DeepSeek-AI, 2024) does not discuss this point explicitly, using low-rank projections (which reduce the parameter count) to shrink the latent cache also opens the possibility of increasing the column dimension of the up-projection matrices to recover those lost parameters. In other words, the model could reallocate the freed parameter budget to add more query heads, potentially preserving model capacity and increasing arithmetic intensity during decoding by sharing a single latent head across a larger number of query heads. As shown in Table 1,

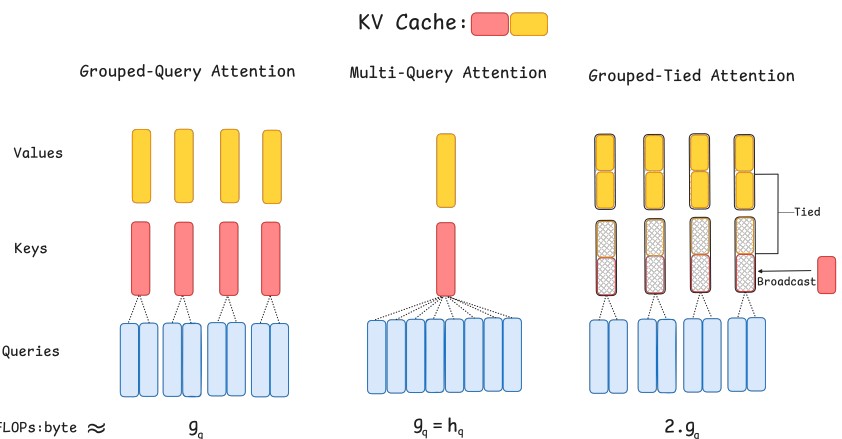

Figure 2: Overview of Grouped-Tied Attention (GTA). A single projection produces a *tied KV* state that serves as both key and value. The full *tied KV* dimension is used as the value. For the keys, half of the key dimension comes from the *tied KV* vector (no positional encoding applied), and the other half comes from a separate single-head projection (where RoPE is applied); this separate half is broadcast to all heads in the group and concatenated with the *tied KV* half. GTA roughly doubles the arithmetic intensity and halves the KV cache size relative to GQA with the same number of groups.

the arithmetic intensity depends on the number of query heads. In general, increasing $g_q$ reduces the KV cache size and increases the arithmetic intensity, effectively maximizing GPU utilization. Let $m_{kv} \in \{1,2\}$ denote the multiplicity of KV, where $m_{kv} = 1$ when $K = V$ (shared states) and $m_{kv} = 2$ when $K \neq V$ (distinct states). In contrast to increasing the group size, $g_q$, raising $m_{kv}$ from 1 to 2 increases the KV cache and reduces the arithmetic intensity.

$$\text{KV}_{\text{Bytes}} = m_{kv} \cdot B \cdot L \cdot \frac{h_q}{g_q} \cdot d_h \times \text{sizeof}(\text{dtype})$$
$$\text{Arithmetic Intensity} \approx \frac{2 \cdot L \cdot h_q}{2 \cdot h_q + \frac{m_{kv} \cdot h_q}{g_q} L} \approx \frac{2 \cdot g_q}{m_{kv}}$$

Here, $B$ is the batch size and $d_h$ is the dimension per head. Moreover, increasing the arithmetic intensity through a larger $g_q$ comes with a trade-off in parallelization capability. This constraint can be quantified by the bounds on $g_q$. Although increasing $g_q$ increases the arithmetic intensity, if $g_q$ grows too large relative to the number of devices, it duplicates the parameters and diminishes parallelization gains. Once the duplication factor $D$ equals $N$, where $N$ is the TP shard count, each machine has a complete copy of the parameters and the KV cache; at that point, the model parallelism does not benefit. For *zero redundancy parallelism*, the number of KV heads $h_{kv} = h_q / g_q$ should be at least $N$ and at most $h_q$, that is, $g_q \leq h_q / N$.

### 3.3 Hardware-Efficient & Parallelizable Attention

#### 3.3.1 Grouped-Tied Attention (GTA)

Singular-value plots reveal a steep decay in the key cache, where almost all variance is captured by a few principal directions, so the keys reside in a low-rank subspace and are highly redundant (Saxena et al., 2024). This effect is even more substantial before applying RoPE, where the keys collapse into an even smaller subspace (Yu et al., 2024a; Sun et al., 2024). Meanwhile, multiple studies show that applying RoPE to a partial slice of head dimension preserves accuracy, so rotating the entire width yields little additional quality (Black et al., 2022; Barbero et al., 2025). If keys are intrinsically low rank and only a slice of each head needs rotation for positional encoding, then rotating every channel and caching the full-rank key tensor wastes memory. Instead, we can rotate just a slice of the head dimension required for positional information; the remaining unrotated channels, which tend to be in low-rank subspace and redundant, can be shared or tied with the value states. GQA already reduces KV cache and memory transfer by letting multiple query heads share one distinct KV head, and it scales efficiently across multiple devices. Building on GQA grouping design, combined

with the above low-rank and partial RoPE insights, we propose *Grouped-Tied Attention* (GTA), which unifies grouping, ties KV to a single state, and partially applies RoPE, all for the purpose of cutting the KV cache size while retaining quality.

In GTA, similar to GQA, each group of query heads shares one distinct KV head. GTA goes further by tying the key and value projection parameters to yield a single state, called the *tied KV*, whose shape matches that of an individual key or value vector (see Figure 2, for an illustration of the differences between the architecture of GTA and GQA). The value path consumes the full dimensionality of the *tied KV* state, whereas the key path reuses only its first half as the unrotated half. The remaining RoPE component of the key comes from a separate one-head projection of the hidden state, broadcast across all groups, and concatenated with the unrotated half to form the full key vector. Empirical ablations show that applying RoPE to the shared half degrades quality even when the rotation is later inverted before reusing this half for the value path, so the tied portion is never rotated. After these steps, the query, key, and value states are defined as follows.

$$Q \in \mathbb{R}^{B \times L \times h_q \times d_h}, \quad \text{KV}, K, V \in \mathbb{R}^{B \times L \times h_{kv} \times d_h}, \quad K_{\text{RoPE}} \in \mathbb{R}^{B \times L \times 1 \times \frac{d_h}{2}}, \quad K_{\text{NoPE}} \in \mathbb{R}^{B \times L \times h_{kv} \times \frac{d_h}{2}}$$

$$K_{\text{NoPE}} = \text{KV}[:,:,:,: \frac{d_h}{2}], \quad V = \text{KV}[:,:,:,:], \quad K = \text{concat}\Big(K_{\text{NoPE}}, \text{broadcast}\big(K_{\text{RoPE}}, h_{kv}\big)\Big)$$

By tying the KV states, we load a single state into on-chip memory, reuse it for both keys and values, and share it across a small set of query heads. This reuse reduces memory transfers, roughly doubles the arithmetic intensity, and halves the KV cache footprint relative to its GQA counterpart with the same number of groups. GQA-4 denotes four distinct key and value heads, whereas GTA-4 denotes four *tied KV* heads. Experiments show that perplexity (see 5.1.1) and performance in downstream tasks (see 5.1.2) remain comparable to its GQA counterpart.

### 3.3.2 *Grouped Latent Attention (GLA)*

MLA's low-rank KV joint compression caches a single latent head of dimension $d_c = 4 d_h$ per token. Because tensor parallelism partitions the key and value up-projections $W^{UK}, W^{UV} \in \mathbb{R}^{(4 d_h \times h d_h)}$ in a column-parallel fashion across ranks, each device must retain the entire latent to reconstruct the keys and values for its heads. Consequently, the latent KV cache is duplicated in every tensor parallel rank, scaling the aggregate KV cache footprint in proportion to the number of ranks. In contrast, GQA stores $2 h_{kv} d_h$ elements per token, with a larger KV cache than MLA when $d_c < 2 h_{kv} d_h$ and $h_{kv} > 2$. However, for a four-degree tensor parallel configuration, MLA and a GQA-8 occupy roughly the same KV cache size per device, although GQA-8 allocates $\sim 4\times$ more bytes to its KV cache size from the base model. The MLA design achieves a high arithmetic intensity, but increasing the arithmetic intensity by sharing latent across all heads prevents shard-wise partitioning (see Section 3.2 for details). Hence, the expected per-device memory reduction from head level parallelism diminishes, limiting parallel efficiency in distributed inference.

We propose Grouped Latent Attention (GLA), which compresses tokens into $h_c$ latent heads, each with dimension $d_c = 2 d_h$ (half of MLA's $4 d_h$). During training, every latent head and its up-projection matrices reconstruct distinct key and value features for the query heads in its group. Consequently, the up-projection matrix for one latent head has column dimension $g_q d_h$ rather than MLA's $h_q d_h$, where $g_q = h_q / h_c$ is the group size or queries per distinct latent head. After weight absorption in decoding, each latent head attends only to the query heads in its group. Sharding the latent heads across tensor parallel ranks provides head-level parallelism without duplicating the latent KV cache when $h_c = \text{TP}$ or reducing it otherwise when $h_c \leq \text{TP}$, thereby enabling efficient distributed scaling.

For concrete exposition, we demonstrate GLA with $h_c = 2$ latent heads, which retains the KV cache size of MLA ($4 d_h$) but half the KV cache per device when $\text{TP} \geq 2$. Setting $\text{TP} = 2$, GLA splits the latent vector into two heads, $c_0^{\text{KV}}$ and $c_1^{\text{KV}}$, and partitions the query heads into two groups. During decoding, each TP rank computes local attention using its assigned latent head and query group, applies its slice of the output projection, and then participates in an AllReduce to sum the partial outputs into the final result. *Formally*:

$$c_{\mathbf{0}}^{\mathrm{KV}}, c_{\mathbf{1}}^{\mathrm{KV}} \in \mathbb{R}^{B \times L \times 2d_h}, \quad Q_{\mathbf{0}}, Q_{\mathbf{1}} \in \mathbb{R}^{B \times 1 \times \frac{h_q}{2} \times (2d_h)}, \quad W_{\mathbf{0}}^{vo}, W_{\mathbf{1}}^{vo} \in \mathbb{R}^{\left(\frac{h_q}{2} \cdot 2d_h\right) \times D},$$

$$O_{\mathbf{0}} = \mathrm{softmax}\left(Q_{\mathbf{0}}(c_{\mathbf{0}}^{\mathrm{KV}})^\top\right)c_{\mathbf{0}}^{\mathrm{KV}}, \quad O_{\mathbf{1}} = \mathrm{softmax}\left(Q_{\mathbf{1}}(c_{\mathbf{1}}^{\mathrm{KV}})^\top\right)c_{\mathbf{1}}^{\mathrm{KV}},$$

$$\tilde{O}_{\mathbf{0}} = O_{\mathbf{0}}W_{\mathbf{0}}^{vo}, \quad \tilde{O}_{\mathbf{1}} = O_{\mathbf{1}}W_{\mathbf{1}}^{vo}, \quad O = \mathrm{AllReduce}\left(\tilde{O}_{\mathbf{0}} + \tilde{O}_{\mathbf{1}}\right).$$

When MLA runs in a TP plus DP hybrid setup to reduce KV cache duplication, uneven sequence lengths can create load imbalance since GPUs that finish short sequences idle until long ones finish processing, hurting latency-sensitive workloads. Small batches below the DP rank also leave many compute units idle. The larger latent head dimension of MLA, $4\,d_h$, can also exhaust the KV cache per device for long sequences or large batches, limiting the batch size or context length relative to GLA. For example, GLA-4 shards the latent into four heads, $h_c = 4$ with $2\,d_h$ per head dimension; with the same configuration (TP=4, DP=2), it halves the cache per device, fetches a smaller cache per step, yet has twice the KV cache size. In our experiments, we set the $h_c = 2$ for GLA, the same KV cache size as MLA ($d_c = 4\,d_h$), where it matches the quality of MLA up to a 1.471 B model (see 5.1 for empirical results) while halving the cache footprint per device when TP $\geq 2$. The smaller cache per device also allows GLA to decrease the DP rank while increasing the TP degree, improving tolerance to workload imbalance.

During decoding, the arithmetic intensity of GLA is $\sim 2 \cdot g_q$ (double of GQA) as shown in Table 1; GLA-2 reaches $\sim h_q$ FLOPs per byte of memory access, similar to MQA in its most aggressive sharing design, but GLA-2 has better quality. Refer to Figure 1 (right), which shows the memory loading schematic of GLA-2 when computing attention. GLA maintains high arithmetic intensity, see Figure 14 (roofline analysis), parallelizes efficiently (see Figure 3 (right)) and provides high model quality (see 5.1). We benchmark the token throughput of GLA in Section 5.2; detailed latency and throughput benchmarks are given in Appendix C.6

## 4 System Optimization: Asynchrony & Distributed Offset Calculation

We describe the system optimization to achieve peak performance for MLA, GTA, and GLA on modern hardware such as H100 GPUs. Thanks to very fast specialized matrix multiplication units such as Tensor Cores, we need careful software pipelining of memory loading and tensor core instructions to always keep the tensor cores busy. This is achieved through the warp specialization technique to exploit asynchrony on modern hardware.

### 4.1 Asynchrony with software pipelining and warp specialization

We use two techniques to overlap compute and memory loading:

1. Software pipelining: we load the next KV block while the current KV block is being used in computation. This classical technique (Lam, 1988) avoids having the tensor cores waiting for memory loading.
2. Warp specialization: we have separate warps performing memory loading with either TMA (tensor memory accelerator) or asynchronous copy (cp.async instruction), and separate warps performing matrix-multiply-accumulate (MMA). The former act as *producer* warps, while the latter act as *consumer* warps (Bauer et al., 2014). This is commonly used in matrix multiplication (Thakkar et al., 2023) and attention (Shah et al., 2024a). This decoupling simplifies the software pipelining, allowing the warp scheduler to overlap the memory loading and compute.

### 4.2 Distributed offset calculation for paged KV

As new attention variants such as MLA, GTA, and GLA stress both the compute and the memory subsystems, one would have to perform memory loading as quickly as possible. Paged KV (Kwon et al., 2023) has become a standard way of storing the KV cache. However, paged KV makes it difficult to use the TMA, a specialized hardware unit that performs address calculation and bound checking to load contiguous blocks of memory. Instead, we use the asynchronous copy instruction (cp.async) where each thread separately issues individual load instructions. The challenge is that address computation is surprisingly expensive,

as it requires 64-bit integer indexing, which translates to multiple instructions per integer multiplication. We instead have multiple threads in the same warp that cooperate to calculate the addresses. As an example, for head dimension 128, to load a block of size 128 x 128 from global memory to shared memory, we use 128 threads, with 16 threads loading per row:

1. Group 128 threads into 8 groups, each consisting of 16 consecutive threads. Each group $g$ for $g = 0,1,...,7$ will be assigned to load rows $g, g+8, ..., g+120$.
2. For each thread $t$, which belongs to the group $g = \lfloor t/16 \rfloor$, read the page index from the page table in row $g + (t \bmod 16) * 8$. Using the page index, compute the global memory address of the paged KV corresponding to this row and store in registers.
3. For row $r$ in $g, g+8, ..., g+120$, all 16 threads $t$ in the same group use warp shuffle to get the global memory address from thread index $g * 16 + (r-g)/8$. Use this address to load the KV cache elements corresponding to this row.

We see that each thread only needs to store the address offset of 1 row (instead of 16 rows), as the address offsets of 16 rows assigned to each group are spread across 16 threads.

This enables high efficiency for arbitrary page size (such as page size 1), where page size 1 suffers no slowdown compared to page size 64, despite a much larger number of address computations. This unlocks use cases such as prefix caching (Kwon et al., 2023) with RadixAttention (Zheng et al., 2024b) which requires page size 1. We benchmark the speed of paged KV with GLA in appendix C.5, showing a 1.2-1.5x speed up.

## 5 Experiments and Results

### 5.1 Model Quality

#### 5.1.1 Validation Perplexity

In addition to reporting validation perplexity on 100M tokens from the FineWeb-Edu validation set, we evaluated perplexity on four additional datasets: Wikipedia and C4 (Colossal Clean Crawled Corpus) portions of RedPajama v1 (Weber et al., 2024), Cosmopedia (Ben Allal et al., 2024) and Pile (Gao et al., 2020), each evaluated using 100M tokens. We also report the average perplexity across these five datasets in Tables 2 and 4. Further validation perplexity results for each dataset are in Appendix C.2.1.

| Method | Small (183M) FineWeb-Edu | Small (183M) Avg. | Medium (433M) FineWeb-Edu | Medium (433M) Avg. | Large (876M) FineWeb-Edu | Large (876M) Avg. |
|---|---|---|---|---|---|---|
| MLA | **16.318** | 40.290 | 12.561 | 28.230 | 11.363 | 24.929 |
| GLA-2 | 16.371 | 40.604 | 12.456 | **27.586** | 11.293 | **24.492** |
| GTA-4 | 16.607 | 42.680 | 12.785 | 29.952 | **11.232** | 24.994 |
| GQA-4 | 16.578 | 42.520 | 12.922 | 30.144 | 11.340 | 25.286 |
| MHA | 16.715 | 41.826 | 12.979 | 29.990 | 11.501 | 25.837 |
| MQA | 16.972 | 43.907 | 13.068 | 30.524 | 11.413 | 25.206 |

Table 2: Validation perplexities (lower is better) on FineWeb-Edu across three model sizes (small, medium, large), along with the average perplexity across five datasets (FineWeb-Edu validation set, Cosmopedia, RPV1 C4, RPV1 Wikipedia, and Pile). The lowest perplexity is in bold, and the second lowest is underlined.

GLA-2 tends to improve upon MLA at medium and large scales, achieving lower validation perplexities on both the FineWeb-Edu set and the five-dataset average while performing on par with MLA in the small model. Notably, GTA addresses key limitations of GQA and further reduces perplexity on medium and large model scales. GLA-2 and $\text{GLA}_q$-2 perform comparably in the large model, achieving the lowest average perplexities of all variants (24.49–24.51) vs. the next-best MLA (24.93). GTA addresses the key limitation of GQA, namely, the inability to reduce the KV cache per device in lower TP degree settings. For the large model, GLA-2 and $\text{GLA}_q$-2 perform similarly on average, demonstrating strong performance compared to the other variants. This trend holds on the scale of the XL (1.47B) model, GTA-4 slightly outperforms GQA-4 in perplexity (10.12 vs. 10.20 on FineWeb-Edu), and GLA maintains an advantage in perplexity over MLA (e.g., 10.21 vs. 10.25 on FineWeb-Edu; see Table 4).

### 5.1.2 Downstream Evaluation

We evaluated zero-shot performance on standard benchmarks: SciQ (Welbl et al., 2017), OpenBookQA (Mihaylov et al., 2018), ARC-Easy subset (Yadav et al., 2019), HellaSwag (Zellers et al., 2019), PIQA (Bisk et al., 2020), WinoGrande (Sakaguchi et al., 2020), and MMLU (Hendrycks et al., 2021).

| Method | Winogrande | SciQ | PiQA | OpenBookQA | MMLU | HellaSwag | Arc Easy | Avg. |
|--------|-----------|------|------|-----------|------|-----------|----------|------|
| **MQA** | 57.1 | 86.0 | 71.3 | 37.8 | 25.7 | 52.2 | 64.7 | 56.4 |
| **GQA-4** | 54.9 | 87.3 | 71.2 | 38.4 | 25.7 | 52.2 | 68.6 | 56.9 |
| **MHA** | 53.8 | 86.0 | 71.4 | 37.4 | 25.2 | 51.5 | 66.7 | 56.0 |
| **GTA-4** | **57.3** | 88.1 | **72.6** | 39.8 | 25.2 | 52.6 | 67.5 | 57.6 |
| **GLA-2** | 55.0 | 88.1 | 72.0 | **40.6** | 25.6 | 52.4 | 69.1 | 57.5 |
| **MLA** | 54.1 | 86.8 | 71.5 | 40.4 | **26.0** | 51.6 | 66.5 | 56.7 |

Table 3: Downstream evaluation for the 876M models (higher is better). $d_R$ denotes the dimension of the RoPE. If not specified for GTA, GLA, and MLA, then the RoPE dimension $d_R$ equals 32.

| Method | FineWeb-Edu PPL | Avg. PPL | Avg. Downstream | KV cache (bytes/token) TP=1 | TP=2 |
|--------|-----------------|----------|-----------------|------------------------------|------|
| MHA | 10.311 | 21.206 | 60.1 | 8192 | 4096 |
| GQA-4 | 10.202 | 21.073 | **60.2** | 2048 | 1024 |
| GTA-4 | **10.129** | **20.823** | **60.2** | **1152** | **640** |
| GLA-2 | 10.218 | 21.163 | 60.0 | **1152** | **640** |
| MLA | 10.256 | 21.199 | 59.1 | **1152** | 1152 |

Table 4: Validation perplexity (lower is better) for the 1.471B model on FineWeb-Edu along with the average perplexity across five datasets. The lowest perplexity is in bold, and the second lowest is underlined. Average downstream evaluation (higher is better) across seven datasets, where the highest accuracy is in bold, and the second highest is underlined. TP refers to the tensor parallelism degree. We report the token size in bytes per device (lower is better) for a single layer. The lowest KV cache size is in bold, and the second lowest is underlined.

Across downstream benchmarks, our proposed attention variants retain or exceed baseline accuracy. For the scale model 876M (listed in Table 3), GLA-2 produces an average accuracy of 57.5%, only 0.1 points below GTA-4 and effectively matches GQA-4, indicating that grouping or tying does not degrade quality. This behavior persists on the 1.471B XL scale, where GLA-2 reaches an average accuracy of 60.0% compared to 59.1% for MLA, while GTA-4 and GQA-4 each record 60.2% (listed in Table 4). These results confirm that our hardware-efficient variants preserve or improve downstream task performance from medium to XL sizes. For more details on downstream evaluation, see Appendix C.2.2. Also, for ablation on small and medium model scales, see Appendix C.3.

### 5.2 Parallelization

GLA scales across GPUs by partitioning latent heads, reducing memory traffic for faster decoding. MLA with tensor parallelism (TP) instead duplicates its single latent head with larger dimension on every GPU; to curb this cost, earlier systems fall back on hybrid tensor and data parallelism (DP) MLA, assigning different batch sequences to separate GPUs, a benefit seen with large batches or high concurrent requests. We validate GLA's scalability on DeepSeek Coder V2 Base (236B parameters, 21B active) quantized to FP8 and served with our FlashAttention 3 kernels using SGLang framework (Zheng et al., 2024b) to benchmark. Experiments compare pure TP on eight H100 GPUs with hybrid TP plus two-way or four-way DP. In the hybrid setup, only the attention submodule is replicated across DP groups; its outputs are all gathered before the MoE feedforward layer and redistributed to mitigate MLA KV cache duplication. GLA configurations with zero redundancy sharding (latent heads evenly distributed across TP ranks) outperform MLA under an equivalent parallel configuration because of the fetching of a smaller latent KV cache per device.

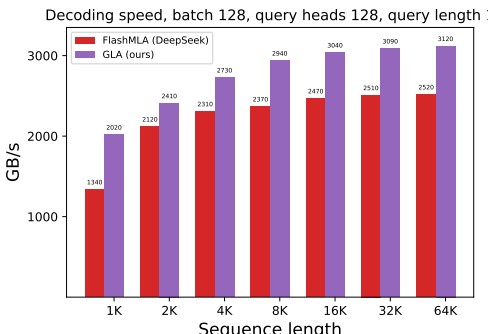 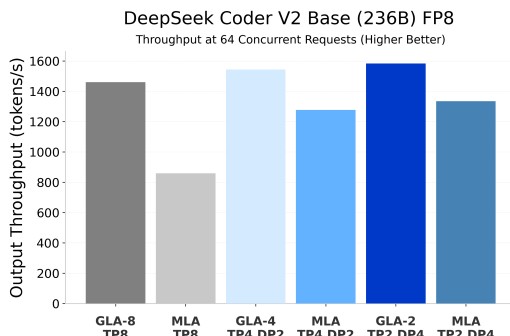

Figure 3: **Left**: Decoding speed of MLA and GLA on H100 80GB SMX5 GPU (theoretical max BF16 compute 989 TFLOPS/s and memory 3350 GB/s), for query length 1 where MLA is close to being bottle-necked by compute (reaching 610 TFLOPS/s) while GLA has not yet saturated compute (360 TFLOPS/s). **Right**: Output throughput (higher better) for 64 concurrent requests for live server benchmark where GLA outperforms MLA under identical parallelism scheme. Also, GLA-8 with pure TP=8 outperforms MLA with a hybrid of TP and DP. The prefill/decode sequence length is 8192/4096 respectively.

Figure 3 (right) shows that with 64 concurrent requests, GLA-8 ($h_c = 8$, $d_c = 256$) on eight GPUs delivers up to $2\times$ the throughput of the single head latent MLA with $d_c = 512$, as it loads a smaller KV cache per device. The advantage persists under hybrid parallelism: GLA-8 with TP=8 outperforms MLA with (TP=2, DP=4), and even with an equivalent hybrid setup of TP and DP, GLA remains ahead when latent heads are sharded without duplication. At the same time, GLA-8 with pure TP=8 can still surpass hybrid TP and DP MLA. The GLA parallelization-friendly design can lift peak throughput and remain resilient to adverse serving loads. Real-world workloads often feature sequence-length imbalance or small batches that trigger straggler effects and idle GPUs.

### 5.3 Speed

We benchmark the decoding kernels for MLA (1 latent head of dimension 512, RoPE dim 64) as shown in Figure 3 left and GLA (2 latent heads of dimension 256 each, RoPE dim 64) as shown in Appendix C.7, Figure 13 (Left), with paged KV with page size 64. We compare our GLA implementation with the most optimized MLA implementation we can find, FlashMLA from DeepSeek (Li, 2025). Our GLA kernel is about 20% faster than FlashMLA in the standard decoding setup (query length 1) and more than $2\times$ faster in the speculative decoding setup (query length 2). Our GLA kernel reaches up to 93% of the maximum memory bandwidth and 70% of the maximum TFLOPS on the H100 GPU, maximizing the use of both the memory and the compute subsystems.

## 6 Conclusion

We demonstrate that focusing on high arithmetic intensity and effective parallelization yields hardware-efficient attention mechanisms while decoding. We introduce Grouped-Tied Attention (GTA), which ties key–value (KV) states, roughly halving the KV cache requirement and doubling the arithmetic intensity relative to GQA with the same number of groups. GTA achieves lower perplexity and better average downstream accuracy than the GQA baseline. We then propose Grouped Latent Attention (GLA), a parallelizable inference-aware attention variant that achieves a high arithmetic intensity and demonstrates decoding speeds up to $2\times$ faster than the baseline DeepSeek FlashMLA (Li, 2025). GLA consistently matches or exceeds MLA accuracy on all model scales and benchmarks evaluated. Moreover, sharding the latent heads across devices, with a head dimension smaller than MLA per device, as a result, GLA fetches a smaller KV cache during decoding, reducing end-to-end latency and increasing throughput in online serving benchmarks by up to $2\times$. Thus, GTA efficiently replaces GQA (requiring only half the KV cache memory). GLA is a practical replacement for MLA due to its scalable partitioning across GPUs and faster decoding speed.

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

# A Discussion

We discuss related work, limitations, and some future directions. For related work, Appendix B reviews the attention mechanisms that reduce the KV cache during pretraining, surveys broader hardware efficient designs, and extends the related work to cover post hoc strategies for faster LLM decoding, studies on applying partial RoPE, and low-rank projections, insights that shaped our methods.

GTA reduces the unsharded KV cache size by roughly half relative to GQA with the same number of KV heads, mainly by caching one tied state per head. With TP $= 8$ and eight KV heads, GTA-8 stores 1.5 $d_h$ per token, where $\frac{d_h}{2}$ of it comes from separate RoPE, while GQA-8 stores 2 $d_h$, a minimal savings of 25%. However, for eight KV heads and with TP $= 4$, the gap widens because GTA-8 allocates 2.5 $d_h$ per token, whereas GQA-8 uses 4 $d_h$, so a moderate TP degree provides greater memory relief. Our GLA experiment utilizes two latent heads to match MLA's unsharded KV cache size, where GLA halves the KV cache per device when TP $\geq 2$ while MLA duplicates the KV cache across devices. Up to the 1.471B scale, model quality remains comparable. As a next step, it is worth evaluating larger-scale models with more latent heads while keeping the head dimension fixed at 2 $d_h$ to further validate GLA's performance. For example, the family of Llama 4 models (up to 400B parameters) employ GQA-8 with $h_q = 40$ and $h_{kv} = 8$, where the KV cache size per token on each device for TP=8 degree is 2 $d_h$, and beyond this degree it would duplicate the cache. A matching GLA-8 with eight latent heads would cache slightly more, 2.5 $d_h$ per token on each device where $\frac{d_h}{2}$ comes from decoupled RoPE; whether GLA-8 improves quality over GQA-8 with roughly similar cache budget is an open question. For half of the KV cache, based on our 1.471B scale experiment, GLA-2 achieves 10.218 validation perplexity versus 10.202 for GQA-4 on FineWeb-Edu. Furthermore, the additional $\frac{d_h}{2}$ per token KV cache footprint of decoupled RoPE can be mitigated by applying RoPE only in partial layers, as shown by Cohere (Yang et al., 2025) and Llama 4 (Meta AI, 2025). We leave these scaling studies for future work.

Section 3.2 shows that the arithmetic intensity scales with the number of query heads. Appendix C.3 presents an ablation that replaces full query and output projections with low-rank versions to cut the parameters. The reduced capacity is offset by widening the column dimension to add additional query heads per group in GQA and GTA. The ablation aimed to improve quality (or preserve quality when the KV cache is further reduced) while boosting the arithmetic intensity, effectively improving GPU utilization. The validation perplexity was slightly worse by 0.1 to 0.2 relative to the baselines when ablating across few ranks and query heads. A detailed study of this trade-off is left for future work. Finally, exploring how parallelization and arithmetic intensity interact in other architectures, such as Mamba (Gu & Dao, 2024; Dao & Gu, 2024) and Linear Attention (Yang et al., 2024), could further exploit modern hardware.

# B Related Work

## B.1 Attention Variants

### B.1.1 Algorithmic

**Pre-training**: Follow-up works to DeepSeek's MLA include Multi-matrix Factorization Attention (MFA) (Hu et al., 2025), which resembles MQA but uses a larger head dimension and factorized query projections, where it shares similar limitations with MQA, particularly regarding inefficient KV cache utilization per device due to duplication and lack of compatibility with tensor parallelism. Tensor Product Attention (TPA) (Zhang et al., 2025) factorizes queries, keys, and values with two rank $R$ projection matrices per state, so the per token cache is $(R_K + R_V)(h + d_h)$ elements. We report validation perplexity in our ablations for TPA in the Appendix C.3. The low-rank structure of TPA supports straightforward head-level TP sharding, but its KV cache scales linearly with $R$ and already exceeds MLA once $R \geq 4$; ranks four and above are important for quality, especially at larger scales, so it can quickly lose the memory-saving advantage.

**Post-hoc adaptation**: In (Lin et al., 2025) they propose SIGMA, which utilizes a novel differentially scaled QKV module specifically optimized and applied during the fine-tuning stage to improve inference efficiency by compressing K while lightly compressing V and

increasing Q. Slim Attention (Graef & Wasielewski, 2025), a post-training approach that keeps only the key vectors and recreates the values on the fly, cutting the context memory for any multi-head attention in half. Our proposed method cuts the KV cache further than both of these approaches. It differs from the techniques above, as it attempts to restructure the attention architecture during pre-training, as opposed to changing the existing attention of the already pre-training model. However, similar distillation methods that have been applied to MLA (Meng et al., 2025; Ji et al., 2025) can be adopted for GLA in the post-training stage to realize the benefits of the low KV cache footprint and easy parallelization.

### B.1.2 Systems

FlashAttention (Dao et al., 2022) reorders attention computation with an I/O-aware tiling strategy that keeps data in high-speed memory, avoiding the need to materialize the entire attention matrix. It drastically reduces memory overhead and yields significant speedups, particularly during decoding with large sequence lengths. FlashAttention-2 (Dao, 2023) further refines the attention kernel by reducing non-matrix multiplication operations and improving parallel work partitioning, delivering additional gains in hardware utilization. Its system-level improvements provide a notable increase in throughput over the original FlashAttention. Then FlashAttention-3 (Shah et al., 2024b) leverages next-generation GPU features, such as asynchronous memory operations and low-precision computation, pushing attention efficiency closer to hardware limits. Natively trainable sparse attention (NSA) (Yuan et al., 2025) introduces hardware-aligned sparse attention with a dynamic hierarchical pattern. This design reduces computational complexity while maintaining near-full attention fidelity, enabling efficient decoding over extremely long sequences. Our proposed methods are orthogonal to these system-level attention optimizations.

### B.2 Additional Approaches to Accelerating Decoding

The following post hoc adaptation design features work in conjunction with the GLA and GTA design architecture, enhancing its performance capabilities.

**Algorithmic**: There have been many algorithmic efforts such as token eviction (Zhang et al., 2023; Xiao et al., 2024) or sharing KV cache between adjacent layers (Brandon et al., 2024), batching to improve GPU utilization (Mukherjee et al., 2023), and speculative decoding (Xia et al., 2023). **Systems**: On the system side, there has been work on quantization (Hooper et al., 2024), CPU offloading (Aminabadi et al., 2022; Sheng et al., 2023; He & Zhai, 2024), and memory management using PagedAttention (Kwon et al., 2023) to mitigate memory fragmentation problems. **Hardware**: In addition, there have been efforts on the hardware side that benefit inference, such as FP4 support (NVIDIA, 2024) or NVLink (NVIDIA Corporation, 2024), to hardware chips designed solely for fast inference (Groq, 2024).

### B.3 Low-Rank Projections

Empirical findings indicate that, before applying RoPE, key activations have a sharply decaying singular-value spectrum (Yu et al., 2024a; Chen et al., 2024), implying that many dimensions contribute minimally. Furthermore, (Singhania et al., 2024) show that keys exhibit substantially reduced intrinsic dimensionality across models, suggesting an inherently low-rank space. For example, (Kobayashi et al., 2024) finds that the regularization of the weight decay drives the combined key-query mapping to an even lower rank. In contrast, value activations exhibit mixed low-rank tendencies. Some studies have shown that cached values do not compress well without a severe accuracy penalty (Chang et al., 2024; Singhania et al., 2024). However, additional findings demonstrate that partial compression can be achieved with acceptable performance degradation (Saxena et al., 2024; Sun et al., 2024). These inconsistencies suggest that the effective rank of values is model- and method-dependent. GTA utilizes the insights from these aforementioned works for its design of tying the key and value within each query group, thereby reducing the KV cache size.

### B.4 Rotary Position Encoding (RoPE)

**Per head dimension** (Barbero et al., 2025) suggests that it may not be necessary to apply RoPE (Su et al., 2023) to every head dimension since the highest frequencies already provide positional discrimination. Other studies (DeepSeek-AI, 2024; Black et al., 2022; Wang &

Komatsuzaki., 2021) find that they can preserve the quality of the model by applying RoPE to only half of the dimension. Inspired by the same insight, GTA rotates only a partial slice of the head dimension and ties the rest to the portion with half of the value head dimension, reducing the KV cache size while preserving accuracy.

**Per layer** (Chen & Yan, 2024) shows that RoPE contributes the most in the early transformer layers, where attention is focused on local syntactic relations. In contrast, the deeper layers shift toward semantic cues, implying diminishing returns from positional rotation later in the stack. (Yang et al., 2025) present RNoPE, a design that interleaves RoPE layers with NoPE layers and limits RoPE to a sliding window, achieving markedly better retrieval at very long context lengths and influencing the architecture choices of Llama 4 (Meta AI, 2025). Overall, applying RoPE to partial layers is beneficial for GLA, considering that the decoupled RoPE, with a single head that is half the head dimension, can be eliminated.

## C Full Experimental Results

### C.1 Experimental Setup

We empirically validate that our simple methods, GTA and GLA, (1) achieve quality comparable to GQA and MLA, (2) are easily parallelizable, and (3) run efficiently on modern hardware such as the H100 GPU. For example, GLA achieves an upstream and downstream quality similar to that of MLA, yet is easier to shard, and our GLA kernel is up to $2\times$ faster than DeepSeek FlashMLA in a speculative decoding setup. $GLA_q$ denotes the configuration in which the query latent is sharded, removing duplication of its down projection across devices and cutting the parameter count per device; we include this variant mainly as an ablation since the query latent is not cached.

We train models on four scales: small (183M), medium (433M), large (876M) and XL (1.471B) parameters on the FineWeb-Edu-100B dataset (Lozhkov et al., 2024), following the configuration of the GPT-3 model (Brown et al., 2020) with the Llama 3 architecture (Grattafiori et al., 2024). The small model is trained on 25 billion tokens, whereas the medium, large, and XL models are each trained on 50 billion tokens. We use the Llama 3 tokenizer (Grattafiori et al., 2024) with a vocabulary size of 128K tokens. We use the AdamW (Loshchilov & Hutter, 2019) optimizer with ($\beta_1, \beta_2$) = (0.9, 0.95), a weight decay of 0.1, and gradient clipping at 1.0. We follow the training recipe from Gu & Dao (2024), using a learning rate scaled by $5\times$ relative to GPT-3 for a model of the same size, with decay of cosine to 1% of the maximum learning rate. We use the configuration of the GPT-3 model (Brown et al., 2020). We first use the configuration of the GPT-3 model for a given parameter size for our MHA baseline, which has the largest parameter budget. Then, we widen the MLPs of every other attention variant until each model matches the MHA parameter count. Essentially, MHA's parameter size is the anchor point.

In GQA-4 & GTA-4, the 4 represents the number of groups or the number of KV heads, $h_{kv} = \frac{h_q}{g_q}$. $GLA_q$ refers to the GLA version in which the latent query is also sharded.

| Model Size | #Param | Micro-batch Size | Batch Size | Learning Rate | #Layer | $d_{model}$ | $h_q$ | $d_h$ |
|---|---|---|---|---|---|---|---|---|
| Small | 183.65M | 16 | 512 | $2.6 \times 10^{-4}$ | 12 | 768 | 12 | 64 |
| Medium | 433.77M | 16 | 512 | $1.45 \times 10^{-4}$ | 24 | 1024 | 16 | 64 |
| Large | 876.55M | 8 | 512 | $1.2 \times 10^{-4}$ | 24 | 1536 | 16 | 96 |
| XL | 1471.12M | 8 | 256 | $1.0 \times 10^{-4}$ | 24 | 2048 | 16 | 128 |

Table 5: Model configuration for the four model sizes in our experiments. We adopted the GPT-3 model configuration, with the Llama 3 architecture as the backbone and its tokenizer as well.

| Method | Model Param | Intermediate size |
|---|---|---|
| MLA | 183.65M | 2128 |
| GLA-2 | 183.51M | 2208 |
| MHA | 183.45M | 2048 |
| MQA | 183.53M | 2520 |
| GTA-4 | 183.40M | 2462 |
| GQA-4 | 183.53M | 2392 |

Table 6: Model parameters and FFN intermediate size for a small model.

| Method | Model Param | Intermediate size |
|---|---|---|
| GLA-2 | 433.89M | 3152 |
| GLA$_q$-2 | 433.89M | 3280 |
| MLA | 433.55M | 3062 |
| GTA-4 | 433.57M | 3320 |
| GQA-4 | 433.77M | 3248 |
| MHA | 433.77M | 2736 |
| MQA | 433.77M | 3376 |

Table 7: Model parameters and FFN intermediate size for a medium model.

| Method | Model Param | Intermediate size |
|---|---|---|
| MHA | 876.55M | 4096 |
| GQA-4 | 876.55M | 4864 |
| MQA | 876.55M | 5056 |
| GTA-4 | 876.55M | 4976 |
| MLA | 876.73M | 4640 |
| MLA ($d_R$ : 48) | 876.74M | 4592 |
| GLA-2 ($d_R$ : 48) | 876.96M | 4914 |
| GLA-2 | 876.73M | 4768 |
| GLA$_q$-2 | 876.44M | 4936 |

Table 8: Model parameters and FFN intermediate size for a large model. $d_R$ denotes the RoPE dimension and the default is 32 for this model size

| Method | Model Param | Intermediate size |
|---|---|---|
| MLA | 1470.58M | 6120 |
| GLA-2 | 1470.78M | 6292 |
| MHA | 1471.12M | 5464 |
| GTA-4 | 1471.22M | 6638 |
| GQA-4 | 1470.83M | 6486 |

Table 9: Model parameters and FFN intermediate size for a XL model.

## C.2 Quality

### C.2.1 Validation Perplexity

| Method | FineWeb-Edu | Cosmopedia | RPV1 C4 | Pile | RPV1 Wikipedia | Avg |
|---|---|---|---|---|---|---|
| MHA | 16.715 | 20.542 | 31.628 | 40.444 | 99.800 | 41.826 |
| GQA-4 | 16.578 | 20.599 | 32.059 | 43.841 | 99.525 | 42.520 |
| MQA | 16.972 | 22.094 | 32.245 | 44.308 | 103.915 | 43.907 |
| GTA-4 | 16.607 | 20.768 | 32.911 | 42.181 | 100.932 | 42.680 |
| GLA-2 | 16.371 | 20.542 | 31.628 | 40.444 | 94.037 | 40.604 |
| GLA$_q$-2 | 16.333 | 20.110 | 31.517 | **38.725** | **92.820** | **39.901** |
| MLA | **16.318** | **20.063** | **31.484** | 39.528 | 94.056 | 40.290 |

Table 10: Validation perplexity for the small model (lower is better). The lowest perplexity is in bold, and the second lowest is underlined. RPV1 refers to RedPajama v1.

| Method | FineWeb-Edu | Cosmopedia | RPV1 C4 | Pile | RPV1 Wikipedia | Avg |
|---|---|---|---|---|---|---|
| GLA-2 | 12.456 | **13.722** | 24.308 | **27.676** | **59.766** | **27.586** |
| GLA$_q$-2 | **12.433** | 13.917 | **24.263** | 28.224 | 60.359 | 27.840 |
| MLA | 12.561 | 14.039 | 24.507 | 28.602 | 61.438 | 28.230 |
| GQA-4 | 12.845 | 14.532 | 25.159 | 30.401 | 65.871 | 29.761 |
| GTA-4 | 12.785 | 14.812 | 25.009 | 30.447 | 66.708 | 29.952 |
| MHA | 12.979 | 14.666 | 25.331 | 30.772 | 66.201 | 29.990 |
| GQA-4 ($qo_R:4\cdot d_h; h_q:48$) | 12.922 | 15.024 | 25.282 | 31.510 | 65.980 | 30.144 |
| MQA | 13.068 | 15.163 | 25.585 | 31.504 | 67.302 | 30.524 |

Table 11: Validation perplexity for the medium model (lower is better). The lowest perplexity is in bold, and the second lowest is underlined. $qo_R$ refers to the rank of the low-rank query and output projections, resulting in reduced model parameters. To offset these lost parameters for fair comparison with the baselines, we increase the query heads $h_q$ to 48. It's beneficial since arithmetic intensity depends on the number of query heads. RPV1 refers to RedPajama v1.

| Method | FineWeb-Edu | Cosmopedia | RPV1 C4 | Pile | RPV1 Wikipedia | Avg |
|---|---|---|---|---|---|---|
| MHA | 11.501 | 12.605 | 22.496 | 27.651 | 54.933 | 25.837 |
| GQA-4 | 11.340 | 12.358 | 22.219 | 26.635 | 53.878 | 25.286 |
| MQA | 11.413 | 12.437 | 22.383 | 26.521 | 53.274 | 25.206 |
| GTA-4 | **11.232** | 12.159 | 22.059 | 26.136 | 53.383 | 24.994 |
| MLA | 11.363 | 12.468 | 22.294 | 25.685 | 52.837 | 24.929 |
| MLA ($d_R:48$) | 11.245 | **12.021** | **22.053** | 25.246 | **52.212** | 24.555 |
| GLA$_q$-2 ($d_R:48$) | 11.337 | 12.144 | 22.234 | **24.620** | 52.612 | 24.589 |
| GLA$_q$-2 | 11.276 | 12.100 | 22.126 | 24.681 | 52.371 | 24.511 |
| GLA-2 | 11.293 | 12.106 | 22.130 | 24.698 | 52.233 | **24.492** |

Table 12: Validation perplexity for the large model (lower is better). $d_R$ refers to the RoPE dimension and the default is 32 for this model size. RPV1 refers to RedPajama v1.

| Method | FineWeb-Edu | Cosmopedia | RPV1 C4 | Pile | RPV1 Wikipedia | Avg |
|---|---|---|---|---|---|---|
| MHA | 10.311 | 10.540 | 20.117 | 22.432 | 42.628 | 21.206 |
| GQA-4 | 10.202 | 10.418 | 19.986 | 22.642 | 42.119 | 21.073 |
| GTA-4 | **10.129** | **10.399** | **19.849** | **22.184** | **41.551** | **20.823** |
| GLA-2 | 10.218 | 10.482 | 20.020 | 22.298 | 42.796 | 21.163 |
| MLA | 10.256 | 10.561 | 20.041 | 22.516 | 42.624 | 21.199 |

Table 13: Validation perplexity for the XL model (lower is better). Bold indicates the lowest score in each column; underlined indicates the second lowest. RPV1 refers to RedPajama v1.

| Method | FineWeb-Edu PPL | Avg PPL | Avg Downstream | KV cache (bytes/token) TP=1 | TP=2 | TP=4 |
|---|---|---|---|---|---|---|
| MHA | 10.311 | 21.206 | 60.1 | 8192 | 4096 | 2048 |
| GQA-4 | 10.202 | 21.073 | **60.2** | 2048 | 1024 | 512 |
| GTA-4 | **10.129** | **20.823** | **60.2** | **1152** | **640** | **384** |
| GLA-2 | 10.218 | 21.163 | 60.0 | **1152** | **640** | 640 |
| MLA | 10.256 | 21.199 | 59.1 | **1152** | 1152 | 1152 |

Table 14: Validation perplexities (lower is better) for the 1.471B model on FineWeb-Edu along with the average perplexity across five datasets (FineWeb-Edu validation, Cosmopedia, RedPajama v1 C4, RedPajama v1 Wikipedia, and Pile). The lowest perplexity is in bold, and the second lowest is underlined. Average Downstream evaluation (higher is better), where the highest accuracy is in bold, the second highest is underlined. TP refers to the tensor parallelism, and we report the KV cache of a token in bytes per device across various TP degrees.

### C.2.2 Downstream Evaluation

| Method | Winogrande | SciQ | PiQA | OpenBookQA | MMLU | HellaSwag | Arc-Easy | Avg |
|---|---|---|---|---|---|---|---|---|
| $GLA_q-2$ | 55.2 | 84.9 | **70.5** | 35.6 | 25.2 | 47.9 | **66.3** | 55.1 |
| $GQA-4_{qo_R}$ | 52.4 | 83.6 | 69.7 | 36.0 | 25.5 | 45.7 | 64.9 | 54.0 |
| GQA-4 | 53.8 | 85.7 | 69.7 | 36.2 | 25.4 | 46.3 | 64.6 | 54.5 |
| GTA-4 | 54.2 | 85.5 | 69.0 | 34.0 | 25.9 | 46.8 | 64.2 | 54.2 |
| MQA | 55.5 | 84.6 | 69.5 | 37.0 | **26.2** | 45.9 | 60.5 | 54.2 |
| GLA-2 | **56.7** | 84.1 | 70.3 | **37.2** | **26.2** | **48.2** | 65.3 | **55.4** |
| MLA | 54.5 | **86.1** | 70.2 | 36.8 | 25.1 | 47.2 | 64.2 | 54.9 |
| MHA | 55.2 | 84.8 | 69.3 | 35.0 | 25.5 | 46.2 | 63.0 | 54.1 |

Table 15: Downstream evaluation for the medium model (higher is better). Bold indicates the highest score in each column; underlined indicates the second highest. $qo_R$ denotes the rank of the low rank query and output projections, set to $4d_h$. To compensate for the reduced parameter count and ensure a fair comparison with the baselines, we increase the number of query heads $h_q$ to 48, which also benefits arithmetic intensity since it scales with the number of query heads.

| Method | Winogrande | SciQ | PiQA | OpenBookQA | MMLU | HellaSwag | Arc-Easy | Avg |
|---|---|---|---|---|---|---|---|---|
| GLA-2 | 57.4 | **91.8** | 73.9 | 40.4 | **26.1** | 58.2 | 72.1 | 60.0 |
| GQA-4 | 59.0 | 91.5 | 74.1 | **41.6** | 25.2 | 58.5 | 71.6 | **60.2** |
| GTA-4 | 58.2 | 91.0 | **75.1** | 40.8 | 25.3 | **58.6** | **72.5** | **60.2** |
| MLA | 56.4 | 89.5 | 73.5 | 39.4 | 25.3 | 58.1 | 71.8 | 59.1 |
| MHA | **60.5** | 90.7 | 73.1 | 41.0 | 25.9 | 57.6 | 71.9 | 60.1 |

Table 16: Downstream evaluation for the XL model (higher is better). Bold indicates the highest score in each column; underlined indicates the second highest.

### C.3 Ablations

Different attention variants reduce the learned parameters and the representational capacity per layer; therefore, these saved parameters need to be redistributed elsewhere. For example, Llama 2 (Touvron et al., 2023) increases the width of the FFNs for MQA and GQA in their ablation to make a fair comparison to MHA. In addition, MQA initially proposed to increase the width to match the parameters to MHA (Shazeer, 2019). Meanwhile, DeepSeek-AI (2024) adjusts the depth of the model, increasing the number of layers for a fair comparison. Altering the depth is less common because it is challenging to make head-to-head comparisons, as there is less flexibility in moderately scaled models to match parameters. (Pope et al., 2022) shrink the head dimension of MHA to match the parameters of MQA and TPA, while (Zhang

et al., 2025) increases the number of query heads to align the parameter count, essentially distributing the saved parameters into the query projections. For instance, in the case of GQA and GTA, the KV heads need to be divisible by the query heads, so there is less flexibility in terms of altering the number of query heads to match as closely as possible to the baseline for fair comparison. In the case of MLA and GLA, increasing the query heads is beneficial since, during decoding, we do not materialize KV. Essentially, a single latent head is shared across all or groups of query heads; therefore, increasing the query heads trivially improves GPU utilization while decoding, as we demonstrated earlier in Table 1 where the arithmetic intensity for MLA and GLA boils down to the number of query heads.

### C.3.1 Ablations: Small Model 188M Parameters

| Attention Type | Model Param | FineWeb-Edu | RPV1 C4 | Cosmopedia | RPV1 Wikipedia | Pile | Avg |
|---|---|---|---|---|---|---|---|
| MHA | 183.45M | 16.71 | 32.24 | 20.54 | 99.79 | 40.44 | 41.94 |
| GQA-4 | 174.01M | 17.07 | 32.93 | 21.62 | 104.1 | 45.09 | 44.16 |
| MQA | 170.47M | 17.39 | 33.70 | 22.53 | 108.0 | 46.79 | 45.68 |
| GTA-4 | 171.95M | 17.04 | 32.95 | 21.82 | 103.5 | 44.67 | 44.00 |
| TPA (r=2) | 172.10M | 17.06 | 32.92 | 21.81 | 99.58 | 44.17 | 43.11 |
| TPA (r=4) | 174.90M | 16.90 | 32.63 | 21.59 | 99.43 | 43.15 | 42.74 |
| MLA | 181.44M | **16.40** | **31.61** | **20.49** | 95.46 | **40.04** | 40.80 |
| $GLA_q$-2 | 175.54M | 16.56 | 31.91 | 20.66 | **94.49** | 40.13 | **40.75** |

Table 17: We ablate by keeping the width of the FFNs (2048) and number of query heads ($h_q$ : 12) constant across the attention variants. Validation perplexity for the small model (lower is better). Bold marks the lowest value in each column, and underlined marks the second-lowest. We include the benchmark for Tensor Product Attention (TPA) (Zhang et al., 2025) with ranks 2 and 4 for the low-rank projection matrices of keys and values. Bold marks the lowest value in each column, and underlined marks the second-lowest. RPV1 refers to RedPajama v1.

| Method | FineWeb-Edu | RPV1 C4 | Cosmopedia | RPV1 Wikipedia | Pile | Avg |
|---|---|---|---|---|---|---|
| $GLA_q$-2 ($h_q$:20) | 16.450 | 31.706 | 20.849 | 95.273 | 40.116 | 40.879 |
| MLA | 16.338 | **31.516** | 20.111 | 94.273 | 39.168 | 40.281 |
| $GLA_q$-2 | **16.337** | 31.517 | **20.110** | 92.820 | **38.726** | **39.902** |
| GTA-4 ($d_R$:16) | 16.870 | 32.732 | 21.089 | 103.508 | 43.103 | 43.460 |
| GTA-4 ($qo_R$:4 $h_q$:24) | 16.517 | 31.946 | 20.727 | 99.962 | 41.462 | 42.123 |
| GTA-4 ($qo_R$:3 $h_q$:36) | 16.496 | 32.048 | 20.537 | 101.030 | 43.787 | 42.780 |
| GQA-4 ($qo_R$:3 $h_q$:36) | 16.546 | 32.048 | 20.786 | 99.684 | 43.787 | 42.570 |
| GQA-4 ($qo_R$:4 $h_q$:24) | 16.405 | 31.754 | 20.530 | 97.979 | 43.302 | 41.994 |

Table 18: The ablations are for different query head counts and projection ranks. Given that arithmetic intensity during decoding depends on the number of query heads, we increase the number of query heads, $h_q$, and the query and output projections are low-rank, denoted by $qo_R$, to compensate for the added parameters. $d_R$ denotes the RoPE dimension. For instance, $d_R = 16$ for GTA-4, we apply RoPE to only 25% of the head dimensions instead of 50% in our proposed approach. Validation perplexity for the small model (lower is better). Bold marks the lowest value in each column, and underlined marks the second-lowest. RPV1 refers to RedPajama v1.

| Method | Model Param | $h_q$ | FineWeb-Edu | RPV1 C4 | Cosmopedia | RPV1 Wikipedia | Pile | Avg |
|---|---|---|---|---|---|---|---|---|
| MHA | 183.45M | 12 | 16.71 | 32.24 | 20.54 | 99.79 | 40.44 | 41.94 |
| MQA | 183.45M | 23 | 17.03 | 32.97 | 21.79 | 104.30 | 44.51 | 44.12 |
| GQA-4 | 183.45M | 20 | 16.79 | 32.44 | 21.19 | 101.50 | 43.05 | 42.99 |
| GTA-4 | 181.40M | 20 | 16.83 | 32.58 | 21.22 | 104.70 | 44.46 | 43.96 |
| GTA-4 | 186.11M | 24 | 16.55 | 32.07 | _20.49_ | 99.13 | 42.78 | 42.20 |
| TPA (r=2) | 183.05M | 21 | 16.74 | 32.33 | 22.32 | 103.60 | 43.32 | 43.66 |
| TPA (r=4) | 183.68M | 19 | 16.75 | 32.32 | 21.63 | 99.78 | 41.81 | 42.46 |
| MLA | 183.02M | 13 | **16.33** | _31.70_ | 20.84 | _95.27_ | **39.16** | _40.66_ |
| GLA$_q$-2 | 183.51M | 13 | _16.44_ | **31.51** | **20.11** | **94.27** | _40.11_ | **40.49** |

Table 19: We ablate by keeping the width of the FFNs (2048) constant across different variants, but increasing the query heads, $h_q$, to match the parameters for fair comparison. Recall that the arithmetic intensity of attention during decoding depends on the number of query heads. Validation perplexity for the small model (lower is better) across different numbers of $h_q$ and identical FFN width. Bold marks the lowest value in each column, and underlined marks the second-lowest.

### C.3.2 Ablations: Medium Model 433M Parameters

In the primary experiment, the medium model (433 M) is trained on 50B tokens, whereas the ablation studies and baseline within this section are trained on 25B tokens.

| Method | Model Param | FineWeb-Edu | RPV1 C4 | Cosmopedia | RPV1 Wikipedia | Pile | Avg |
|---|---|---|---|---|---|---|---|
| GTA-4 | 433.57M | 13.250 | 25.804 | 15.051 | 70.733 | 31.091 | 31.19 |
| MLA | 433.55M | 13.066 | 25.367 | 14.515 | _66.667_ | 30.027 | 29.93 |
| GLA-2 | 433.60M | _12.985_ | _25.216_ | **14.422** | **65.730** | **29.515** | **29.57** |
| GLA$_q$-2 | 433.89M | **12.957** | **25.108** | _14.434_ | 67.182 | 29.909 | _29.92_ |
| MLA | 433.55M | 13.087 | 25.411 | 14.582 | 66.847 | _29.725_ | 29.93 |
| GQA-4 | 433.77M | 13.395 | 25.999 | 15.211 | 70.403 | 31.650 | 31.33 |
| MHA | 433.77M | 13.552 | 26.286 | 15.330 | 72.488 | 32.124 | 31.96 |
| MQA | 433.77M | 13.574 | 26.436 | 15.615 | 72.789 | 33.513 | 32.39 |
| TPA (r=2) | 433.77M | 13.186 | 25.612 | 15.186 | 68.709 | 31.269 | 30.79 |
| TPA (r=4) | 433.96M | 13.143 | 25.538 | 14.672 | 66.877 | 30.396 | 30.12 |

Table 20: The width of the FFN is modified to match parameters as closely as possible across variants. They are all trained on 25B tokens. Validation perplexity for the medium model (lower is better). Bold marks the lowest value in each column, and underlined marks the second-lowest. We benchmark TPA using low rank key and value projection matrices at ranks 2 and 4. RPV1 refers to RedPajama v1.

| Method | Model Param | Winogrande | SciQ | PiQA | OpenBook QA | MMLU | HellaSwag | Arc Easy | Avg |
|---|---|---|---|---|---|---|---|---|---|
| TPA (r=2) | 433.77M | 53.8 | 84.1 | 68.3 | 36.3 | 25.8 | 45.4 | 63.5 | 53.8 |
| TPA (r=4) | 433.96M | 51.8 | 83.7 | 68.6 | 35.4 | 25.2 | 45.5 | 65.7 | 53.6 |
| GTA-4 | 433.57M | 55.2 | 84.3 | 69.2 | 34.9 | **26.0** | 45.3 | 63.1 | 54.0 |
| MLA | 433.55M | 54.0 | 82.9 | 69.3 | **39.9** | 25.4 | 46.1 | 65.4 | **54.7** |
| GLA-2 | 433.60M | **55.5** | 83.8 | **70.0** | 35.2 | 25.5 | 46.2 | **66.6** | 54.6 |
| GLA$_q$-2 | 433.89M | 54.3 | **85.9** | 69.4 | 37.2 | 24.9 | **46.3** | 63.8 | 54.5 |
| MLA | 433.55M | 53.2 | 83.9 | 69.3 | **39.9** | 25.4 | 45.9 | 64.3 | 54.5 |
| GQA-4 | 433.77M | 55.0 | 82.8 | 69.2 | 34.5 | 25.3 | 45.0 | 63.6 | 53.6 |
| MHA | 433.77M | 51.7 | 85.5 | 69.3 | 35.6 | 25.4 | 44.2 | 62.8 | 53.5 |
| MQA | 433.77M | 51.5 | 83.7 | 68.3 | 37.4 | 25.7 | 44.4 | 62.6 | 53.3 |

Table 21: The width of the FFN is modified to match parameters as closely as possible across variants. All models are trained on 25 B tokens. Downstream evaluation for the medium model (higher is better). Bold indicates the highest score in each column; underlined indicates the second highest. We benchmark TPA using low-rank key and value projection matrices at ranks 2 and 4. RPV1 refers to RedPajama v1.

| Method | Model Param | $h_q$ | FineWeb-Edu PPL | Avg PPL | Avg Downstream | KV Cache (bytes/token) TP=1 | TP=2 |
|---|---|---|---|---|---|---|---|
| GLA-2 | 434.73M | 26 | **13.236** | **30.358** | 53.9 | 576 | 320 |
| MQA | 433.77M | 31 | 13.703 | 33.022 | 53.4 | 256 | 256 |
| GQA-4 | 433.77M | 28 | 13.567 | 32.019 | 52.6 | 1024 | 512 |
| GTA-4 | 428.26M | 28 | 13.401 | 31.475 | 53.6 | 576 | 320 |
| GLA$_q$-2 | 434.76M | 32 | 13.321 | 30.909 | 53.4 | 576 | 320 |
| MLA | 434.32M | 23 | 13.249 | 30.875 | **54.2** | 576 | 576 |
| MHA | 433.77M | 16 | 13.552 | 31.956 | 53.5 | 4096 | 2048 |
| TPA(r=2) | 433.47M | 29 | 13.367 | 31.171 | 53.2 | 744 | 624 |
| TPA(r=4) | 432.59M | 26 | 13.404 | 31.030 | 54.0 | 1440 | 1232 |

Table 22: We run ablation by keeping the width of the FFNs (2736) constant across different variants but increasing the query heads, $h_q$, to match the parameters for fair comparison. Recall that the arithmetic intensity of attention during decoding depends on the number of query heads. We report the validation perplexity (lower is better) for FineWeb-Edu, along with the average perplexity across five datasets: FineWeb-Edu validation set, Cosmopedia, RedPajama v1 C4, RedPajama v1 Wikipedia, and Pile. The lowest perplexity is in bold, and the second lowest is underlined. We report the average downstream evaluation (higher is better), where the highest accuracy is in bold, and the second-highest is underlined. TP refers to the tensor parallelism, and we report the KV cache of a token in bytes per device across various TP degrees. We benchmark TPA using low-rank key and value projection matrices at ranks 2 and 4. RPV1 refers to RedPajama v1.

| Method | Model Param | $h_q$ | FineWeb-Edu | RPV1 C4 | Cosmopedia | RPV1 Wikipedia | Pile | Avg |
|---|---|---|---|---|---|---|---|---|
| GLA-2 | 434.73M | 26 | **13.236** | **25.710** | **14.665** | **67.764** | **30.416** | **30.358** |
| MQA | 433.77M | 31 | 13.703 | 26.721 | 15.732 | 75.651 | 33.305 | 33.022 |
| GQA-4 | 433.77M | 28 | 13.567 | 26.341 | 15.514 | 72.141 | 32.534 | 32.019 |
| GTA-4 | 428.26M | 28 | 13.401 | 26.036 | 15.217 | 71.095 | 31.628 | 31.475 |
| GLA$_q$-2 | 434.76M | 32 | 13.321 | 25.859 | 15.085 | 69.604 | 30.677 | 30.909 |
| MLA | 434.32M | 23 | 13.249 | 25.734 | 15.089 | 69.569 | 30.735 | 30.875 |
| MHA | 433.77M | 16 | 13.552 | 26.286 | 15.330 | 72.488 | 32.124 | 31.956 |
| TPA(r=2) | 433.47M | 29 | 13.367 | 25.936 | 15.322 | 70.091 | 31.138 | 31.171 |
| TPA(r=4) | 432.59M | 26 | 13.404 | 26.059 | 15.409 | 69.805 | 30.473 | 31.030 |

Table 23: We ablate by keeping the width of the FFNs (2736) constant across different variants but increasing the number of query heads, $h_q$, to match the parameters for fair comparison. Recall that the arithmetic intensity of attention during decoding depends on the number of query heads. Validation perplexity for the medium model (lower is better). Bold marks the lowest value in each column, and underlined marks the second-lowest. There is less flexibility for GQA and GTA to match parameters since the number of KV heads $h_{kv}$ needs to be divisible by the $h_q$. We benchmark TPA using low-rank key and value projection matrices at ranks 2 and 4. RPV1 refers to RedPajama v1.

| Method | Model Param | Winogrande | SciQ | PiQA | OpenBook QA | MMLU | HellaSwag | Arc Easy | Avg |
|---|---|---|---|---|---|---|---|---|---|
| GLA | 434.73M | 52.5 | 84.1 | 69.2 | 36.3 | 25.6 | **45.4** | 64.3 | 53.9 |
| MQA | 433.77M | **54.6** | 83.9 | 67.9 | 34.7 | 25.8 | 43.7 | 63.6 | 53.4 |
| GQA-4 | 433.77M | 52.2 | 82.3 | 68.0 | 34.9 | 24.9 | 43.9 | 62.6 | 52.6 |
| GTA-4 | 428.26M | 53.4 | 85.1 | 68.2 | 34.9 | **25.9** | 44.8 | 63.6 | 53.6 |
| GLA$_q$-2 | 434.76M | 53.2 | 83.7 | 68.4 | 35.2 | 25.1 | 44.8 | 63.6 | 53.4 |
| MLA | 434.32M | 53.9 | **85.6** | **69.5** | 35.4 | 24.4 | 44.9 | **65.9** | **54.2** |
| MHA | 433.77M | 51.7 | 85.5 | 69.3 | **35.6** | 25.4 | 44.2 | 62.8 | 53.5 |
| TPA(r=2) | 433.47M | 51.9 | 84.3 | 68.9 | 35.0 | 25.1 | 44.7 | 62.1 | 53.2 |
| TPA(r=4) | 432.59M | 52.9 | 83.3 | 68.4 | 38.2 | 25.8 | 44.7 | 65.1 | 54.0 |

Table 24: We run ablations by keeping the FFN width (2736) constant across variants while increasing the number of query heads $h_q$ to match parameters for fair comparison. The arithmetic intensity of attention during decoding depends on $h_q$. Downstream evaluation for the medium model (higher is better). Bold indicates the highest score in each column; underlined indicates the second-highest. GQA and GTA have less flexibility because $h_{kv}$ must divide $h_q$. TPA is benchmarked with low-rank key and value projections at ranks 2 and 4. RPV1 = RedPajama v1.

| Method | Model Param | Winogrande | SciQ | PiQA | OpenBook QA | MMLU | HellaSwag | Arc Easy | Avg |
|---|---|---|---|---|---|---|---|---|---|
| GLA | 434.73M | 52.5 | 84.1 | 69.2 | 36.3 | 25.6 | **45.4** | 64.3 | 53.9 |
| MQA | 433.77M | **54.6** | 83.9 | 67.9 | 34.7 | 25.8 | 43.7 | 63.6 | 53.4 |
| GQA-4 | 433.77M | 52.2 | 82.3 | 68.0 | 34.9 | 24.9 | 43.9 | 62.6 | 52.6 |
| GTA-4 | 428.26M | 53.4 | 85.1 | 68.2 | 34.9 | **25.9** | 44.8 | 63.6 | 53.6 |
| GLA$_q$-2 | 434.76M | 53.2 | 83.7 | 68.4 | 35.2 | 25.1 | 44.8 | 63.6 | 53.4 |
| MLA | 434.32M | 53.9 | **85.6** | **69.5** | 35.4 | 24.4 | 44.9 | **65.9** | **54.2** |
| MHA | 433.77M | 51.7 | 85.5 | 69.3 | 35.6 | 25.4 | 44.2 | 62.8 | 53.5 |
| TPA(r=2) | 433.47M | 51.9 | 84.3 | 68.9 | 35.0 | 25.1 | 44.7 | 62.1 | 53.2 |
| TPA(r=4) | 432.59M | 52.9 | 83.3 | 68.4 | **38.2** | 25.8 | 44.7 | 65.1 | 54.0 |

Table 25: We run ablations by keeping the FFN width (2736) constant across variants while increasing the number of query heads $h_q$ to match parameters for fair comparison. The arithmetic intensity of attention during decoding depends on $h_q$. Downstream evaluation for the medium model (higher is better). Bold indicates the highest score in each column; underlined indicates the second-highest. GQA and GTA have less flexibility because $h_{kv}$ must divide $h_q$. TPA is benchmarked with low-rank key and value projections at ranks 2 and 4. RPV1 = RedPajama v1.

## C.4 Per Token KV Cache Size per Device

| Method | KV cache per Token | KV cache per token per Device (2 GPUs) | KV cache per token per Device (4 GPUs) | KV cache per token per Device (8 GPUs) |
|---|---|---|---|---|
| MHA | $64d_h$ | $32d_h$ | $16d_h$ | $8d_h$ |
| GQA-4 | $16d_h$ | $8d_h$ | $4d_h$ | $2d_h$ |
| MQA | $2d_h$ | $2d_h$ | $2d_h$ | $2d_h$ |
| MLA | $4.5d_h$ | $4.5d_h$ | $4.5d_h$ | $4.5d_h$ |
| GLA-2 | $4.5d_h$ | $2.5d_h$ | $2.5d_h$ | $2.5d_h$ |
| GTA-4 | $8.5d_h$ | $4.5d_h$ | $2.5d_h$ | $1.5d_h$ |

Table 26: An example of KV cache per token for llama 3 8B model configuration with $h_q$ : 32 and $h_{kv}$ : 8 across various TP degrees. $d_h$ denotes head dimension.

## C.5 Speed

We show here that our technique (distributed offset calculation) significantly speeds up the attention kernel when using paged KV. Typically, the attention kernel speed slows down when the page size is small since there is more overhead of address calculation (Kwon et al., 2023). However, a smaller page size reduces fragmentation and unlocks new use cases, such as prefix caching (Zheng et al., 2024b). We benchmark the speed of the decoding kernels for GLA (2 latent heads of dimension 256 each, RoPE dimension 64) with paged KV, as shown in Figure 4. We compare page size 1 and page size 64, with or without distributed offset calculation. With distributed offset calculation, page size 1 does not suffer from the slowdown, matching the speed of page size 64. On the other hand, without distributed offset calculation, page size 1 is $1.3\times$ slower than page size 64. We see that the distributed offset calculation gives a speedup of $1.2\times$ for page size 64 and a speedup of $1.5\times$ for page size 1.

## C.6 End-to-End Latency and Throughput

**Live server setup.** We use SGLANG, a production-oriented service framework, and run every experiment in its live server mode, ensuring that HTTP parsing, dynamic queueing, and GPU kernel invocation are timed together (Zheng et al., 2024a). Evaluating in this mode exposes the queueing overhead and network latency absent from offline testing, thus revealing how GLA and MLA behave under real deployment constraints. The load generator sends 1280 prompts with a chosen concurrency limit, which controls the number of active requests at once. The server combines these active requests into small batches on the fly, so the limit affects load pressure rather than the fixed batch size. We used the pre-trained weights of the DeepSeek-Coder-V2 Base (236B parameters with 21B active parameters),

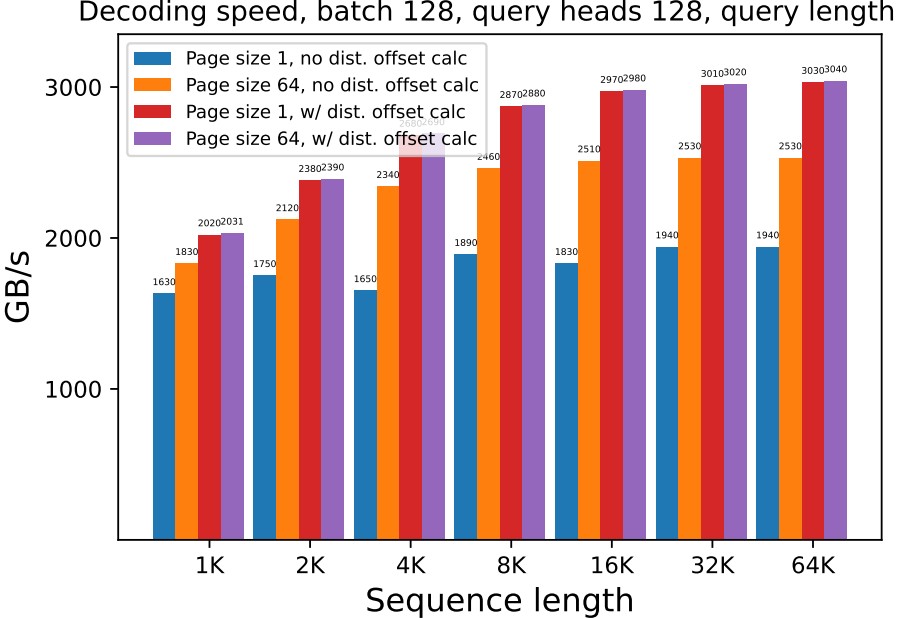

Figure 4: Decoding speed of GLA on H100 80GB SMX5 GPU (theoretical max BF16 compute 989 TFLOPS/s and memory 3350 GB/s), for query length 2, with BF16 format. Distributed offset calculation gives 1.2-1.5x speedup, allowing page size 1 to match the speed of page size 64.

quantized to FP8, and served with our FlashAttention3 kernels. For the benchmarks, we set the page size to 64. To simulate GLA, we restructure the MLA latent dimension to GLA with randomly initializing weights since we benchmark performance, not accuracy, in this phase. We also employ chunked-prefills (Agrawal et al., 2023), with a tile length of 8192 tokens, and run the prefill kernel one block at a time. Decode batches are formed independently, so prefill tokens never mix with decode tokens by default.

**Parallelism and metrics.** Every transformer block is sharded across eight GPUs with tensor parallel, while the MoEs feedforward layers are further partitioned by expert parallel. We also benchmark a mix of data parallelism and tensor parallelism, and whenever data parallelism is enabled, only the attention submodule is replicated across data parallel groups. Its outputs are all-gathered before the MoEs feed-forward layer, then redistributed to mitigate the KV cache duplication of MLA. We benchmark a broad spectrum of inference workloads to assess GLA and MLA under both identical parallelism configurations and in cases where GLA employs only tensor parallelism, while MLA combines tensor and data parallelism. We report four service-level metrics: end-to-end (E2E) latency, time-to-first token (TTFT), inter-token latency (ITL), and output throughput. All values in the figures are summarized by their median, which is less sensitive to heavy-tail behavior in large-scale interactive systems. Additionally, we provide the mean values in the tables.

### C.6.1 *Tensor Parallelism: GLA vs. MLA*

In this configuration with TP degree 8 across x8 H100 GPUs, GLA-8 employs eight latent heads, where each token has to cache a latent dimension of 256, whereas MLA maintains a 512-dimensional latent cache duplicated across devices. Both methods have decoupled the RoPE dimension of 64. Figures 5 and Table 27 reveal consistent gains for GLA-8 at every load level. With 16 concurrent requests, GLA-8 reduces the median end-to-end latency from 136 to 117 seconds, a reduction of approximately 15%, while increasing token throughput by approximately 17%. When the concurrency limit rises to 64, GLA-8 completes in 179 seconds compared to 381 seconds for MLA, cutting latency by 53% percent; the first token now arrives after 12 seconds rather than about 3 minutes, and throughput grows by about 70% to 1461

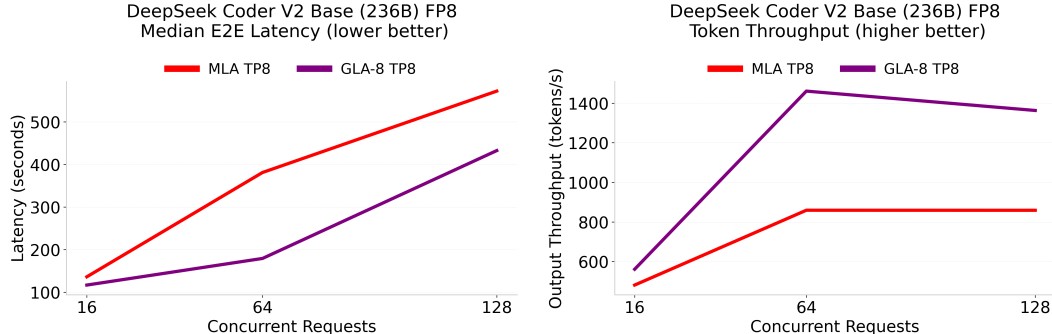

Figure 5: Median end-to-end latency (left), lower is better, and output throughput (right), higher is better, of MLA and GLA-8 under pure TP service on eight GPUs. Prefill/Decode are fixed at 8 K/4 K tokens as concurrency is swept over 16, 64, 128. Because GLA-8 stores roughly half the KV-cache per token under a TP degree of 8, it fetches less data during decoding and consistently outperforms MLA.

tokens per second. Even with 128 concurrent requests, GLA-8 still reduces latency by around 24% and maintains a throughput lead of nearly 60%. These advantages stem from the smaller KV cache footprint of GLA-8 per device, which reduces memory traffic, allows more active requests to fit on the GPUs, and shortens the waiting time before computation can begin.

| Method | Prefill/Decode length | Max conc. /#Prompts | Median E2E Latency (s) | Median TTFT (s) | Median ITL (ms) | Output Throughput (token/s) |
|---|---|---|---|---|---|---|
| GLA-8 *(TP8)* | 8K/4K | 16/1280 | 116.83 | 3.74 | 27.10 | 561.06 |
| MLA *(TP8)* | 8K/4K | 16/1280 | 136.23 | 3.98 | 31.77 | 481.09 |
| GLA-8 *(TP8)* | 8K/4K | 64/1280 | 179.32 | 11.96 | 38.16 | 1460.61 |
| MLA *(TP8)* | 8K/4K | 64/1280 | 381.13 | 192.70 | 43.03 | 858.95 |
| GLA-8 *(TP8)* | 8K/4K | 128/1280 | 432.54 | 223.09 | 45.99 | 1362.84 |
| MLA *(TP8)* | 8K/4K | 128/1280 | 572.20 | 392.07 | 43.04 | 858.69 |

Table 27: Median service-level metrics for MLA and GLA on x8 GPU TP server; the table reports end-to-end latency, time to first token, inter-token latency, and output throughput at concurrency limits of 16, 64, and 128. GLA surpasses MLA on every measure, cutting latency by more than half and lifting throughput by roughly 70% at the mid-load point of 64 concurrent requests.

| Method | Prefill/Decode length | Max conc. /#Prompts | Mean E2E Latency (s) | Mean TTFT (s) | Mean ITL (ms) | Output Throughput (token/s) |
|---|---|---|---|---|---|---|
| GLA-8 *(TP8)* | 8K/4K | 16/1280 | 116.80 | 3.59 | 27.64 | 561.06 |
| MLA *(TP8)* | 8K/4K | 16/1280 | 136.21 | 3.83 | 32.32 | 481.09 |
| GLA-8 *(TP8)* | 8K/4K | 64/1280 | 179.45 | 11.94 | 40.90 | 1460.61 |
| MLA *(TP8)* | 8K/4K | 64/1280 | 301.57 | 118.99 | 44.58 | 858.95 |
| GLA-8 *(TP8)* | 8K/4K | 128/1280 | 370.62 | 168.84 | 49.27 | 1362.84 |
| MLA *(TP8)* | 8K/4K | 128/1280 | 589.07 | 406.52 | 44.58 | 858.69 |

Table 28: Mean service-level metrics for MLA and GLA-8 on x8 GPU TP server; the table reports end-to-end latency, time to the first token, inter-token latency, and output throughput at concurrency limits of 16, 64, and 128

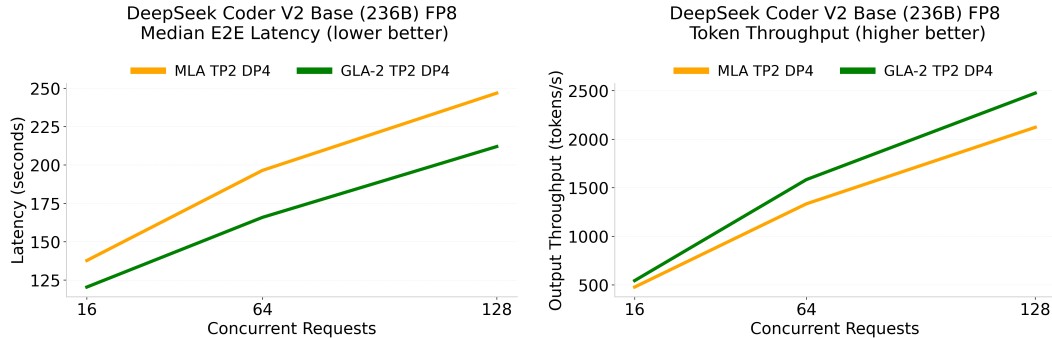

Figure 6: Median end-to-end latency (left), lower is better, and output throughput (right), higher is better, of MLA and GLA under expert parallelism across 8 GPUs with 4 DP groups solely for attention. Prefill/Decode are fixed at 8K/4K tokens as concurrency is swept over 16, 64, 128. GLA outperforms MLA consistently under various concurrencies.

### C.6.2   Data Parallelism + Tensor Parallelism: GLA vs. MLA

Figures 6 and Table 29 demonstrate how the balance between compute capacity and memory traffic changes when parallel data attention is introduced. Under mixed TP 2 + DP 4, as shown in Figures 6 and Table 29, GLA-2 with two latent heads each with dimension 256, shortens the median end-to-end latency from 137 to 120 seconds and increases throughput from 477 tokens per second to 544 tokens per second at a light load of 16 concurrent requests. At 64 concurrency, the advantage grows to roughly 16% lower latency (196 s relative to 166 s) and 19% higher throughput of 1334 tokens per second vs. 1584 tokens per second. Similarly, in mixed TP 4 + DP 2, as shown in Figures 7 and Table 31, GLA-4 consistently outperforms MLA under various concurrent requests.

However, when the limit reaches 128 requests, as shown in Figures 8 and Figures 9, MLA with the hybrid of TP with degree 2 and DP with degree 4, overtakes GLA-8 in pure TP with degree 8 by using the extra replicas to spread the batch and saturate all compute units; MLA now delivers about 56% more tokens per second (2122 tokens per second vs. 1363 tokens per second) and finishes roughly 43% earlier (247 seconds relative to 433 seconds). The cross-over occurs because the added compute lanes of data parallelism offset its cache duplication overhead once the server is heavily loaded. In contrast, GLA-8 in pure TP has already reached the memory bandwidth ceiling and cannot scale further, demonstrating that data parallelism is useful only at large concurrency.

| Method | Prefill/Decode length | Max conc. /#Prompts | Median E2E Latency (s) | Median TTFT (s) | Median ITL (ms) | Output Throughput (token/s) |
|---|---|---|---|---|---|---|
| GLA-2 *(TP2, DP4)* | 8K/4K | 16/1280 | 120.43 | 5.56 | 27.95 | 543.77 |
| MLA *(TP2, DP4)* | 8K/4K | 16/1280 | 137.33 | 5.92 | 31.97 | 477.30 |
| GLA-2 *(TP2, DP4)* | 8K/4K | 64/1280 | 165.86 | 14.12 | 35.01 | 1583.51 |
| MLA *(TP2, DP4)* | 8K/4K | 64/1280 | 196.47 | 14.78 | 42.35 | 1334.18 |
| GLA-2 *(TP2, DP4)* | 8K/4K | 128/1280 | 211.98 | 25.32 | 40.90 | 2474.20 |
| MLA *(TP2, DP4)* | 8K/4K | 128/1280 | 246.81 | 26.93 | 49.12 | 2121.88 |

Table 29: Median service-level results for GLA-2 and MLA when both run with eight-way tensor parallelism and four-way data parallel attention. GLA-2 shows lower latency and higher throughput at the two lighter loads; at the heaviest load, MLA narrows the gap, but GLA-2 still leads by about 14% on latency (lower is better) and throughput (higher is better).

| Method | Prefill/Decode length | Max conc. /#Prompts | Mean E2E Latency (s) | Mean TTFT (s) | Mean ITL (ms) | Output Throughput (token/s) |
|---|---|---|---|---|---|---|
| GLA-2 *(TP2, DP4)* | 8K/4K | 16/1280 | 120.51 | 5.18 | 28.16 | 543.77 |
| MLA *(TP2, DP4)* | 8K/4K | 16/1280 | 137.29 | 5.50 | 32.18 | 477.30 |
| GLA-2 *(TP2, DP4)* | 8K/4K | 64/1280 | 165.52 | 14.20 | 36.95 | 1583.51 |
| MLA *(TP2, DP4)* | 8K/4K | 64/1280 | 196.46 | 14.76 | 44.37 | 1334.18 |
| GLA-2 *(TP2, DP4)* | 8K/4K | 128/1280 | 211.86 | 25.39 | 45.53 | 2474.20 |
| MLA *(TP2, DP4)* | 8K/4K | 128/1280 | 247.04 | 26.57 | 53.84 | 2121.88 |

Table 30: Mean service-level results for GLA-2 and MLA when both run with eight-way tensor parallelism and four-way data parallel attention. GLA-2 shows lower latency under various metrics.

| Method | Prefill/Decode length | Max conc. /#Prompts | Median E2E Latency (s) | Median TTFT (s) | Median ITL (ms) | Output Throughput (token/s) |
|---|---|---|---|---|---|---|
| GLA-4 *(TP4, DP2)* | 8K/4K | 16/1280 | 118.34 | 4.51 | 27.48 | 553.29 |
| MLA *(TP4, DP2)* | 8K/4K | 16/1280 | 135.86 | 4.71 | 31.66 | 482.42 |
| GLA-4 *(TP4, DP2)* | 8K/4K | 64/1280 | 170.66 | 12.80 | 36.07 | 1542.96 |
| MLA *(TP4, DP2)* | 8K/4K | 64/1280 | 205.39 | 13.36 | 44.51 | 1276.25 |
| GLA-4 *(TP4, DP2)* | 8K/4K | 128/1280 | 222.36 | 23.87 | 43.36 | 2357.85 |
| MLA *(TP4, DP2)* | 8K/4K | 128/1280 | 462.03 | 237.35 | 49.66 | 1341.89 |

Table 31: Median service-level results for GLA-4 and MLA when both run with eight-way tensor parallelism and four-way data parallel attention. GLA-4 shows slightly lower latency (lower is better) and higher throughput (higher is better) at the two lighter concurrent requests; at the heaviest load GLA-4 performs significantly better than MLA.

| Method | Prefill/Decode length | Max conc. /#Prompts | Mean E2E Latency (s) | Mean TTFT (s) | Mean ITL (ms) | Output Throughput (token/s) |
|---|---|---|---|---|---|---|
| GLA-4 *(TP4, DP2)* | 8K/4K | 16/1280 | 118.44 | 4.21 | 27.89 | 553.29 |
| MLA *(TP4, DP2)* | 8K/4K | 16/1280 | 135.84 | 4.44 | 32.09 | 482.42 |
| GLA-4 *(TP4, DP2)* | 8K/4K | 64/1280 | 169.87 | 12.77 | 38.36 | 1542.96 |
| MLA *(TP4, DP2)* | 8K/4K | 64/1280 | 205.37 | 13.38 | 46.88 | 1276.25 |
| GLA-4 *(TP4, DP2)* | 8K/4K | 128/1280 | 222.30 | 23.87 | 48.46 | 2357.85 |
| MLA *(TP4, DP2)* | 8K/4K | 128/1280 | 380.39 | 165.82 | 52.40 | 1341.89 |

Table 32: Mean service-level results for GLA-4 and MLA when both run with eight-way tensor parallelism and four-way data parallel attention.

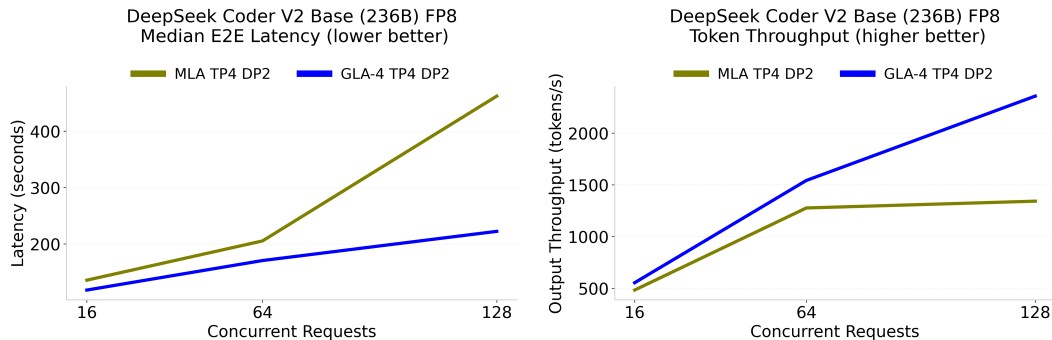

Figure 7: Mean service-level metrics for MLA and GLA on x8 GPU TP server; the table reports end-to-end latency, time to first token, inter-token latency, and output throughput at concurrency limits of 16, 64, and 128 with fixed prefill/decode length of 8K/4K

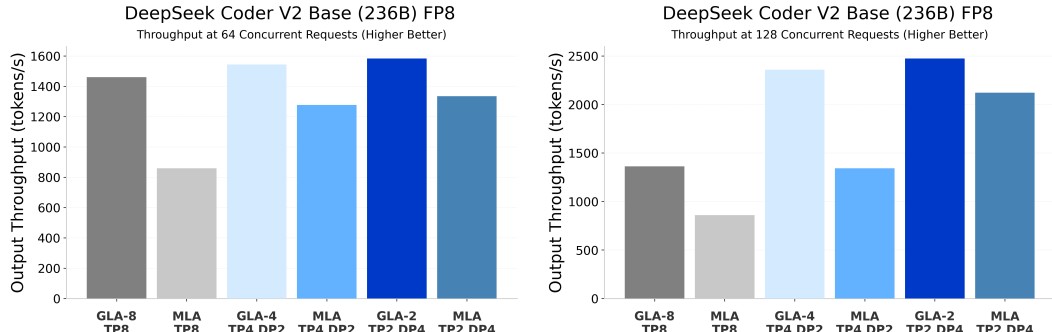

Figure 8: Token throughput at 64 concurrent requests (left) and 128 concurrent requests (right), where higher is better. The prefill/decode sequence length is 8192/4096. GLA outperforms MLA under equivalent parallelism configurations, but MLA with a hybrid of TP and DP at the 128 concurrent requests has higher throughput than GLA under pure TP.

| Method | Prefill/Decode length | Max conc. /#Prompts | Median E2E Latency (s) | Median TTFT (s) | Median ITL (ms) | Output Throughput (token/s) |
|---|---|---|---|---|---|---|
| GLA-2 *(TP8)* | 32K/4K | 16/1280 | 166.18 | 18.11 | 32.47 | 394.76 |
| MLA *(TP2, DP4)* | 32K/4K | 16/1280 | 188.37 | 36.13 | 34.02 | 347.88 |
| GLA-2 *(TP8)* | 64K/4K | 16/1280 | 219.90 | 61.94 | 35.70 | 224.29 |
| MLA *(TP2, DP4)* | 64K/4K | 16/1280 | 313.68 | 118.37 | 37.00 | 208.59 |

Table 33: Median service-level results for GLA-2 only with eight-way tensor parallelism and MLA under mix parallelism scheme with eight-way tensor parallelism and four-way data parallel attention. GLA-2 has 14% higher throughput (higher is better) relative to MLA for prefill length of 32K while 7% higher throughput for 64K prefill length.

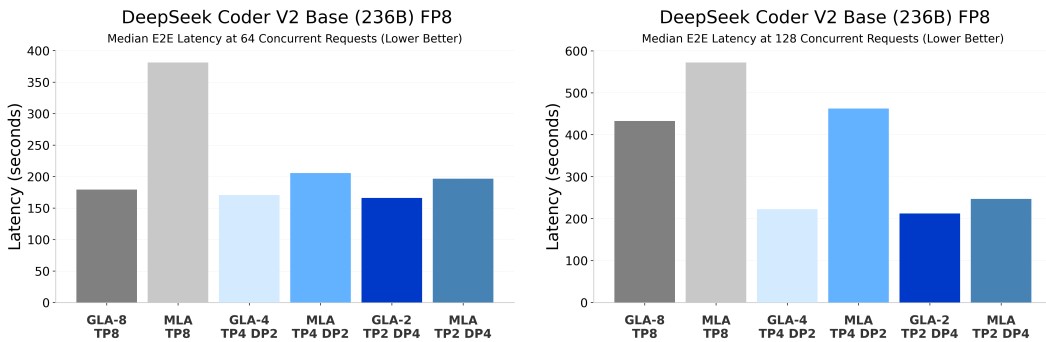

Figure 9: Median E2E latency at 64 concurrent request (left) and 128 concurrent request (right), where higher is better. The prefill/decode sequence length is 8192/4096. GLA outperforms MLA under equivalent parallelism configurations, but MLA with a hybrid of TP and DP at the 128 concurrent requests has higher throughput than GLA-8 under pure TP.

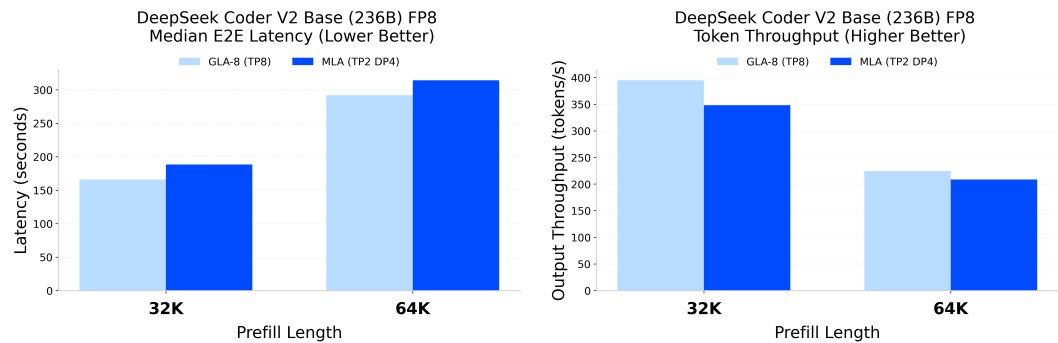

Figure 10: Median end-to-end latency (left), lower is better, and output throughput (right), higher is better, under TP and DP groups solely for attention and expert parallelism for GLA and MLA respectively, across 8 GPUs. Demonstrating long-context with a moderately high number of concurrent requests.

| Method | Prefill/Decode length | Max conc. /#Prompts | Mean E2E Latency (s) | Mean TTFT (s) | Mean ITL (ms) | Output Throughput (token/s) |
|---|---|---|---|---|---|---|
| GLA-2 *(TP8)* | 32K/4K | 16/1280 | 166.00 | 18.09 | 36.12 | 394.76 |
| MLA *(TP2, DP4)* | 32K/4K | 16/1280 | 188.37 | 31.25 | 38.36 | 347.88 |
| GLA-2 *(TP8)* | 64K/4K | 16/1280 | 291.90 | 112.39 | 43.63 | 224.29 |
| MLA *(TP2, DP4)* | 64K/4K | 16/1280 | 314.16 | 102.16 | 51.77 | 208.59 |

Table 34: Mean service-level results for GLA-2 only with eight-way tensor parallelism and MLA under mix parallelism scheme with eight-way tensor parallelism and four-way data parallel attention.

### C.6.3 Data Parallelism: Workload Imbalance

The random ratio parameter is a fraction of the minimum length that the benchmarks' random-request generator may assign to any individual prefill or decode sequence. For example, with a random ratio of 0.125 and a sequence length of 4096 tokens, each request is created with lengths drawn uniformly from the integer range of 512 to 4096 tokens, giving every batch a consistent lower bound while retaining a realistic spread of sequence sizes. The random ratio is applied to prefill sequence lengths (131K) and decode (4K). The experiments

in this section demonstrate workload imbalance with varying sequence lengths across the batch, which can leave GPUs idle.

In Figure 11 and Table 35, we demonstrate where for a long prefill of 131K and a relatively long decode of 4K, where the sequence length is uniformly sampled within the batch, GLA-8 with TP degree 8 has about 2.7× higher throughput than MLA in hybrid TP with degree 2 in four data parallel ranks. Because every NCCL collective in a data-parallel group must be entered by all ranks, one replica that is still busy with a very long sequence forces every other replica, and its tensor-parallel shards, to wait, so throughput collapses to the speed of that single straggler, with pure TP-8, there is no extra data-parallel barrier, so only the eight shards that hold the weights pause for one another; a long sequence slows that shard group, but leaves the rest of the cluster working, keeping GPU utilization much higher.

| Method | Prefill/Decode length | Rand. ratio | Max conc. /#Prompts | Median E2E Latency (s) | Median TTFT (s) | Median ITL (ms) | Output Throughput (token/s) |
|---|---|---|---|---|---|---|---|
| GLA-8 *(TP8)* | 131K/4K | 0 | 4/1280 | 80.21 | 6.42 | 25.10 | 101.59 |
| MLA *(TP2, DP4)* | 131K/4K | 0 | 4/1280 | 203.04 | 32.03 | 28.64 | 37.50 |
| GLA-8 *(TP8)* | 131K/4K | 0.125 | 4/1280 | 89.57 | 7.58 | 25.45 | 100.68 |
| MLA *(TP2, DP4)* | 131K/4K | 0.125 | 4/1280 | 233.69 | 38.54 | 28.66 | 37.20 |
| GLA-8 *(TP8)* | 32K/4K | 0.125 | 4/1280 | 55.97 | 1.14 | 22.32 | 165.78 |
| MLA *(TP2, DP4)* | 32K/4K | 0.125 | 4/1280 | 73.82 | 3.29 | 25.73 | 125.31 |

Table 35: With a *random ratio of 0*, each request chooses its prefill and decode lengths uniformly from a single token up to the maximum lengths. With a *random ratio of 0.125*, the lengths are sampled uniformly, but now the range starts at 12.5% of the maximum specified length. GLA-8 with pure TP has higher throughput and lower median end-to-end latency than the hybrid TP + DP MLA configuration across both long and moderate context settings.

| Method | Prefill/Decode length | Rand. ratio | Max conc. /#Prompts | Mean E2E Latency (s) | Mean TTFT (s) | Mean ITL (ms) | Output Throughput (token/s) |
|---|---|---|---|---|---|---|---|
| GLA-8 *(TP8)* | 131K/4K | 0 | 4/1280 | 80.93 | 7.78 | 35.54 | 101.59 |
| MLA *(TP2, DP4)* | 131K/4K | 0 | 4/1280 | 219.43 | 41.82 | 86.31 | 37.50 |
| GLA-8 *(TP8)* | 131K/4K | 0.125 | 4/1280 | 91.73 | 8.72 | 35.93 | 100.68 |
| MLA *(TP2, DP4)* | 131K/4K | 0.125 | 4/1280 | 248.35 | 46.89 | 87.21 | 37.20 |
| GLA-8 *(TP8)* | 32K/4K | 0.125 | 4/1280 | 55.66 | 1.19 | 23.59 | 165.78 |
| MLA *(TP2, DP4)* | 32K/4K | 0.125 | 4/1280 | 73.65 | 3.79 | 30.26 | 125.31 |

Table 36: With a random ratio of 0, each request draws its prefill and decode lengths uniformly from one token up to the maximum lengths. With a random ratio of 0.125, the range begins at 12.5% of the maximum length. Across both long and moderate context settings, GLA-8 in pure tensor parallel form sustains higher throughput and lower mean end-to-end latency than MLA that combines tensor and data parallelism.

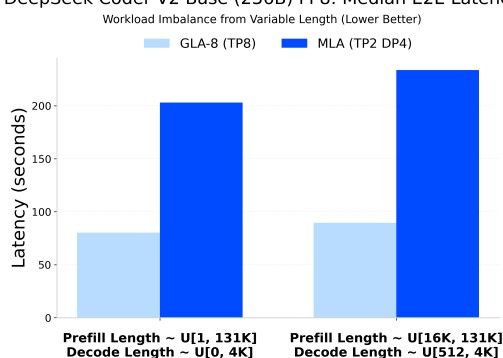 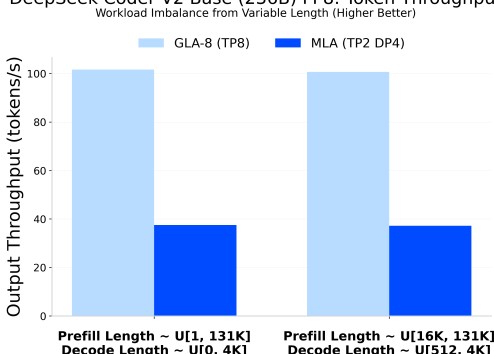

Figure 11: Median end-to-end latency (left), lower is better, and output throughput (right), higher is better, where the sequence length can vary and it is sampled from a uniform distribution. GLA using pure TP outperforms MLA with hybrid TP and DP.

| Method | Prefill/Decode length | Rand. ratio | Max conc. /#Prompts | p99 E2E Latency (s) | p99 TTFT (s) | p95 ITL (ms) | Output Throughput (token/s) |
|---|---|---|---|---|---|---|---|
| GLA-8 *(TP8)* | 131K/4K | 0 | 4/1280 | 175.47 | 19.91 | 26.78 | 101.59 |
| MLA *(TP2, DP4)* | 131K/4K | 0 | 4/1280 | 566.48 | 117.44 | 30.80 | 37.50 |
| GLA-8 *(TP8)* | 131K/4K | 0.125 | 4/1280 | 182.06 | 19.98 | 26.98 | 100.68 |
| MLA *(TP2, DP4)* | 131K/4K | 0.125 | 4/1280 | 572.05 | 119.69 | 30.77 | 37.20 |
| GLA-8 *(TP8)* | 32K/4K | 0.125 | 4/1280 | 99.08 | 2.49 | 23.49 | 125.31 |
| MLA *(TP2, DP4)* | 32K/4K | 0.125 | 4/1280 | 135.87 | 8.61 | 27.48 | 165.78 |

Table 37: For ninety-ninth percentile values of latency, TTFT, and ITL (lower is better), GLA-8 with pure tensor parallel remains faster for the extreme long-context workload, while MLA with hybrid parallelism shows higher output throughput in the moderate context run.

### C.6.4 Latency Sensitive Workloads

In latency-sensitive workloads, the predominant objective is to minimize end-to-end response time, particularly time to first token, to meet strict service level objectives rather than to maximize aggregate throughput. Because a larger batch can increase the queueing and prefill delay, latency-sensitive serving keeps the batch size very small, at the expense of throughput to deliver faster responses. In Table 38, GLA-8 with pure TP at eight degrees manages to reduce latency by x2 and cut the time to first token by almost x4 relative to MLA with a hybrid of TP and DP, where it is necessary to mitigate the duplication of the KV cache.

| Method | Prefill/Decode length | Max conc. /#Prompts | Median E2E Latency (s) | Median TTFT (s) | Median ITL (ms) | Output Throughput (token/s) |
|---|---|---|---|---|---|---|
| GLA-8 *(TP8)* | 64K/256 | 3/1280 | 24.60 | 12.96 | 24.54 | 31.17 |
| MLA *(TP8, DP4)* | 64K/256 | 3/1280 | 54.25 | 46.76 | 28.14 | 14.14 |

Table 38: Under latency-sensitive scenarios, GLA with only tensor parallelism outperforms MLA with a mix of TP and DP solely for attention for long context short decode scenarios by over 50% for both end-to-end median latency (lower is better) and output throughput (higher is better).

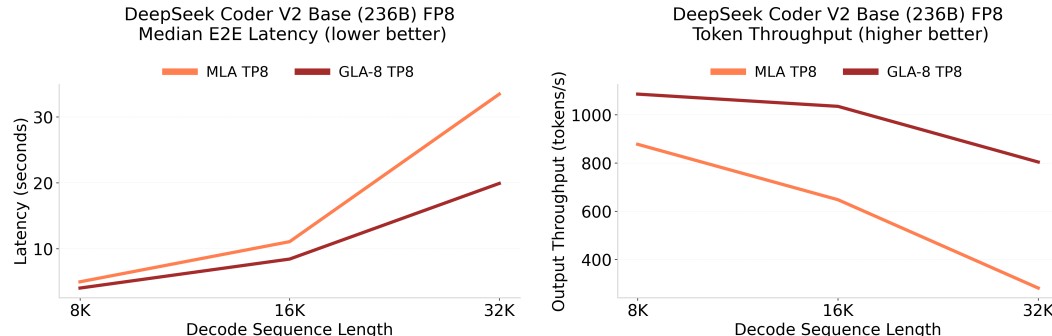

Figure 12: Demonstration of MLA and GLA for TP for degree of 8 on long decode tasks. With 256 number of prompts and 32 concurrent requests across various decode sequence lengths with fixed 2K prefill sequence length.

| Method | Prefill/Decode length | Max conc. /#Prompts | Mean E2E Latency (s) | Mean TTFT (s) | Mean ITL (ms) | Output Throughput (token/s) |
|---|---|---|---|---|---|---|
| GLA-8 *(TP8)* | 64K/256 | 3/1280 | 24.62 | 12.76 | 46.51 | 31.17 |
| MLA *(TP2, DP4)* | 64K/256 | 3/1280 | 54.26 | 46.47 | 30.55 | 14.14 |

Table 39: Mean service-level results for GLA and MLA. GLA shows lower latency (lower is better) than MLA under various metrics.

### C.6.5 Decode Heavy Workloads

In decode-heavy workloads, the generated continuation is so long that the sequential decode phase dominates wall-clock time, resulting in latency and memory bandwidth for the KV cache being the primary bottleneck. Since the model will be performing sequential decoding most of the time, batching offers minimal benefit. In Figure 12, where there is a short prefill of 256 and long decoding of up to 32K, with GLA-8 and MLA across eight-degree parallelism, GLA-8 can generate up to 2.5x higher throughput.

### C.6.6 Small Context and Short Chat

Small Context describes inference requests in which both the prompt and the generated continuation are very short relative to the model context window. For example, a voice assistant answers a brief query in a single response. In Table 40, GLA-8 with eight latent heads in eight-degree parallelism has a lower latency relative to MLA with hybrid TP with degree 2 and DP with four data-parallel ranks since it is a single batch setting, the GPUs in the three out of four DP rank parallel groups remain idle, and GLA-8 has to fetch half the KV cache per layer; therefore, it outperforms MLA.

| Method | Prefill/Decode length | Max conc. /#Prompts | Median E2E Latency (s) | Median TTFT (s) | Median ITL (ms) | Output Throughput (token/s) |
|---|---|---|---|---|---|---|
| GLA-8 *(TP8)* | 256/128 | 1/1280 | 2.49 | 0.11 | 18.72 | 51.45 |
| MLA *(TP2, DP4)* | 256/128 | 1/1280 | 2.91 | 0.12 | 21.94 | 43.96 |

Table 40: Under short chat scenario where there is usually one concurrent request, GLA with only tensor parallelism has 17% higher throughput (higher is better) than MLA with mix of tensor parallelism and data parallelism

| Method | Prefill/Decode length | Max conc. /#Prompts | Mean E2E Latency (s) | Mean TTFT (s) | Mean ITL (ms) | Output Throughput (token/s) |
|---|---|---|---|---|---|---|
| GLA-8 *(TP8)* | 256/128 | 1/1280 | 2.49 | 0.11 | 18.73 | 51.45 |
| MLA *(TP2, DP4)* | 256/128 | 1/1280 | 2.91 | 0.12 | 21.95 | 43.96 |

Table 41: Mean service-level results for GLA and MLA. GLA shows lower latency (lower is better) than MLA under various metrics

| Method | Prefill/Decode length | Max conc. /#Prompts | Median E2E Latency (s) | Median TTFT (s) | Median ITL (ms) | Output Throughput (token/s) |
|---|---|---|---|---|---|---|
| GLA-8 *(TP8)* | 2K/2K | 8/1280 | 47.18 | 0.86 | 22.54 | 346.92 |
| MLA *(TP2, DP4)* | 2K/2K | 8/1280 | 56.35 | 0.82 | 27.04 | 290.91 |

Table 42: For moderate size prefill and decode sequence lengths with moderate number of concurrent requests, GLA with only tensor parallelism has roughly 19% higher throughput (higher is better) than MLA with mix of tensor parallelism and data parallelism.

| Method | Prefill/Decode length | Max conc. /#Prompts | Mean E2E Latency (s) | Mean TTFT (s) | Mean ITL (ms) | Output Throughput (token/s) |
|---|---|---|---|---|---|---|
| MLA *(TP2, DP4)* | 2K/2K | 8/1280 | 56.37 | 0.81 | 27.12 | 290.91 |
| GLA-8 *(TP8)* | 2K/2K | 8/1280 | 47.22 | 0.82 | 22.67 | 346.92 |

Table 43: Mean service-level results for GLA and MLA. GLA shows lower latency (lower is better) than MLA under various metrics

## C.7   Kernel Execution Time

We benchmark the latency of the attention kernels in these two settings on H100 GPUs (ignoring communication overhead) in Tables 44 and 45. GLA with TP = 2 can be 1.3-1.5 times faster than MLA with DP in these settings.

| Seqlen | MLA (DP) | GLA (TP=2) |
|---|---|---|
| 2048 | 15.0 $\mu$s | 16.1 $\mu$s |
| 8192 | 20.8 $\mu$s | 19.1 $\mu$s |
| 32768 | 35.9 $\mu$s | 27.6 $\mu$s |
| 131072 | 81.0 $\mu$s | 55.0 $\mu$s |

Table 44: Attention kernel latency ($\mu$s) for MLA vs. GLA on two GPUs with batch=1

| Seqlens in batch | MLA (DP) | GLA (TP=2) |
|---|---|---|
| [1024]*15+[8192] | 23.8 $\mu$s | 25.4 $\mu$s |
| [1024]*15+[16384] | 29.8 $\mu$s | 26.2 $\mu$s |
| [1024]*15+[32768] | 41.1 $\mu$s | 30.6 $\mu$s |
| [1024]*15+[65536] | 56.0 $\mu$s | 42.6 $\mu$s |

Table 45: Attention kernel latency ($\mu$s) with 2 H100 GPUs (8B model), imbalanced workload

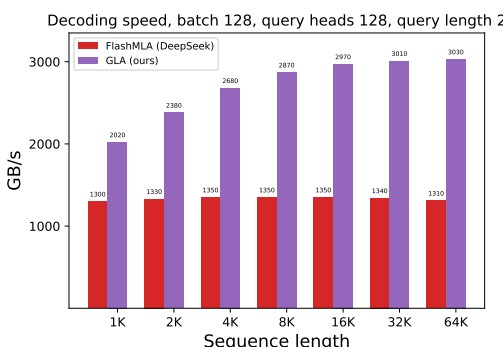
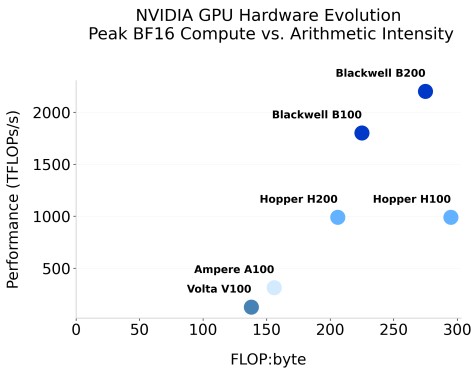

Figure 13: **Left**: Decoding speed of MLA and GLA on H100 80GB SMX5 GPU (theoretical max BF16 compute 989 TFLOPS/s and memory 3350 GB/s), for query length 2. At query length 2, GLA saturates compute (700 TFLOPS/s) and memory (3030 GB/s). **Right**: Peak BF16 theoretical peak FLOPs (TFLOPS/s) versus the arithmetic intensity for successive NVIDIA GPUs (Volta V100 with FP16). Performance computing has historically grown faster than bandwidth, with the H100 architecture (NVIDIA, 2022), which has the most drastic FLOPs-to-byte ratio increase relative to its predecessor. The decoding workload lies far left, so every device, even Blackwell B200, stays memory-bound and reaches only a few percent of its nominal TFLOP rate.

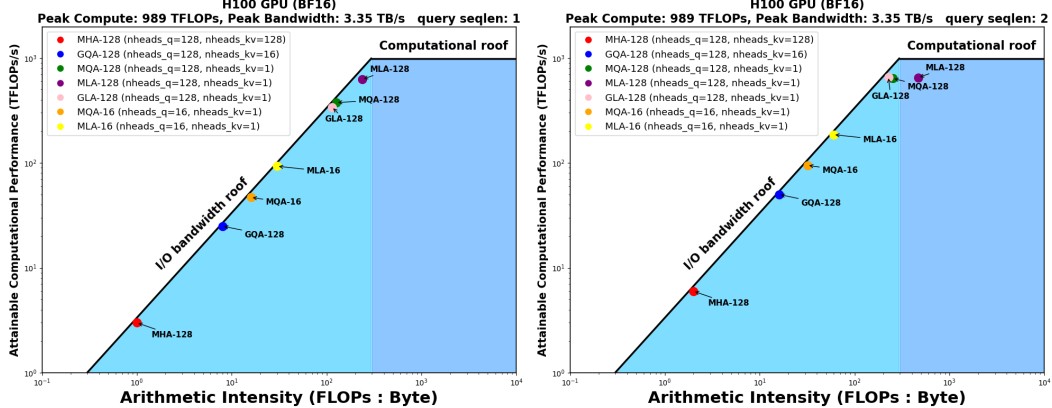

Figure 14: Roofline analysis of BF16 decoding on a single H100 80GB SXM5. In this figure only, the numeric suffix (e.g., GQA-*128*) indicates the number of query heads $h_q$; elsewhere in the paper, it denotes $h_{kv}$. **Left,** $L_q$=1**:** With $h_q$=128, MLA attains an arithmetic intensity of $\sim 2 \cdot h_q$=256, near the compute roof of $\sim$ 295 FLOPs/byte of H100, whereas GLA–128 with two latent heads remains on the I/O roof with arithmetic intensity of $\sim h_q$=128 similar to MQA. **Right,** $L_q$=2**:** e.g., in speculative decoding setting when query length is 2, for $h_q$ : 128 climbs beyond the roof and becomes compute bound, while GLA with two latent heads, sits at the inflection point, and can run up to 2$\times$ faster.

