# OpenReview forum: "Hardware-Efficient Attention for Fast Decoding"
_colmweb.org/COLM/2025/Conference — COLM 2025_

### Official Review · Reviewer_TDTP · 2025-05-10

**Rating:** 6
**Confidence:** 4
**Ethics Flag:** 1

**Summary:**

In this work, the authors analyze the attention design from the perspective of arithmetic intensity and parallelization, and propose two novel attention variants named GTA and GLA, respectively. GTA ties the KV states onto a single projection, and GLA brings parallelization-aware modification to MLA. The experiments demonstrate the efficacy of the two variants on both quality and efficiency.

**Questions To Authors:**

1. In GTA, How is $K_{rope}$ paralleled across multiple GPUs?
2. What is the relationship between the design principles between GTA and GLA?  It seems they are totally independent.
3. In Section 5.2, it is better to also consider attention-specialized parallelization methods like sequence parallel (SP).
4. Does the observed speedup in Figure 3 originate from low-level optimizations,  considering the uniform sequence lengths?

**Reasons To Accept:**

1. A timely topic concerning LLM inference efficiency.
2. A new perspective. The authors theoretically analyze the impact of various attention variants on computational intensity and parallelization patterns.
3. Sufficient quality experiments. The authors conduct comprehensive experiments on multiple datasets to demonstrate that the generation quality is well preserved.

**Reasons To Reject:**

1. The main logic is not so clear. This paper attempts to analyze attention variant design from a hardware-efficiency perspective. However,  while Section 3 demonstrates certain trade-offs, it fails to establish clear design principles. Notably, increased computational intensity or modified parallelization schemes do not inherently guarantee acceleration.
2. Lack of algorithm-level insights. The paper's primary contribution lies in its reformulation of the attention computation's mathematical expression, which necessitates deeper algorithmic insights. The design of GTA references certain data characteristics,  but most are derived from prior works. In designing GLA, the authors directly presents the modified computation scheme without justifying how the model quality is preserved.
3. The acceleration of GLA is not sufficiently general, as it is constrained to specific scenarios and inputs.
4. Writing issues. The variable definitions appear too late in the manuscript, and under-sized fonts in Figure 2 affect reading.

---

> ### Author Response · Authors · 2025-06-03
> **Official Comment by Authors**
>
> We thank **TDTP** for the constructive feedback, noting that our work offers “a new perspective” through “analysis of how different attention variants affect computational intensity and parallelization” and for recognizing that our methods “demonstrate the efficacy of the two variants on both quality and efficiency” as shown by “sufficient quality experiments.”  We also acknowledge their concern that GLA acceleration may be scenario-specific. In the rebuttal, we conducted broader benchmarks confirming GLA’s efficacy and added clearer notation, along with a detailed rationale for the design decisions of both attention variants.
>
> >In regards to the logic and design principles and "increased computational intensity or modified parallelization schemes do not inherently guarantee acceleration"
>
> Loading KV cache dominates latency (large-batch & long-context). On H100 (BF16), the roofline is $\sim295$ FLOPs/byte, MHA attains $\sim1$, so approximately $250\times$ more (add-multiply) per byte loaded can be performed without impacting latency. As **TDTP** mentioned, increasing intensity or modified parallelization do not inherently guarantee acceleration, but our cost model for $\sigma(QK^\top)V$ shows that by tying & grouping it can shrink the KV cache along the $(g_q,m_{kv})$ plane up to $g_q \le \frac{h_q}{N}$. Within the bound, raising $g_q$ boosts GPU utilization while reducing KV cache which proportionally lowers wall-clock time until the roofline is reached or the cache is duplicated per device, where no speedup is gained. The design principle is that the bound turns into a rule-driven checklist such that tying parameters while moderately increasing $g_q$ w.r.t to TP shard count can improve throughput & latency.
>
> >What is the relationship between the design principles between GTA and GLA? It seems they are totally independent.
>
> Both GLA & GTA follow the same design principle to cut active KV bytes to boost arithmetic intensity, occupying different points on the same $(g_q,m_{kv})$ plane & complementing each other. GTA pushes toward $m_{kv}=1$ with fixed $g_q$. GLA keeps $m_{kv}=1$ but pushes $g_q$ toward the zero duplication limit $g_q \le \frac{h_q}{N}$; where each latent head ($2\ d_h$) span across the TP ranks and each head reconstructs unique KV head for query heads in its group.
>
> > In regards to the design of GTA
>
> Singular value plots show keys span a low-rank subspace concentrated in few directions; without RoPE, the subspace shrinks further [1,2 ,3]. Rotating only part of each head maintains accuracy, so full-width rotation yields little benefit [4]. Caching these full-rank rotated keys wastes memory, whereas rotating the needed slice and sharing the rest with values removes this overhead.
>
>
> >In regards to the design GLA
>
> The intuition here is that we can maintain the same model capacity (total KV cache size) by doubling the number of heads and reducing the size of each head. This design choice makes it easy to shard, leading to faster kernels and end-to-end inference time, while maintaining model quality (our eval in Sec 5)
>
> >The acceleration of GLA is not sufficiently general, as it is constrained to specific scenarios and inputs. Does the observed speedup in Figure 3 originate from low-level optimizations, considering the uniform sequence lengths? In Section 5.2, it is better to also consider attention-specialized parallelization methods like sequence parallel (SP).
>
> We provide highly detailed live benchmark results for various metrics (latency, throughput, TTFT, ITL, TFTT) across different workloads (varying sequence lengths, long-decode short prefill, large-batch throughput, small-batch long-context, etc.) under different parallelism schemes, benchmarking MLA vs. GLA in the general response & anonymous link for visualization. SP is almost always used in conjunction with TP in a hybrid setup, so regardless of whether TP is involved, GLA is trying to avoid or mitigate duplication of the KV cache relative to MLA. By grouping, it reduces the KV cache per device by half.
>
> >Writing issues. The variable definitions appear too late in the manuscript, and under-sized fonts in Figure 2 affect reading.
>
> We have restructured the methodology section extensively (3.2, 3.3.1 & 3.3.2) to rename notations and explain more meticulously the design decisions and algorithmic explanation for GLA & GTA.
>
> >In GTA, How is  K_rope  paralleled across multiple GPUs?
>
> With this moderate scale setup, K_rope uses one head at half the usual head dimension. Larger models that require TP should assign K_rope multiple heads to avoid per-device duplication. Future work should test whether a multi-head K-rope with a 25% $d_h$ surpasses the current configuration.
>
> [1] https://arxiv.org/abs/2408.05646
>
> [2] https://arxiv.org/abs/2406.07056
>
> [3] https://arxiv.org/abs/2410.21465
>
> [4] https://arxiv.org/abs/2410.06205

---

> > ### Comment · Reviewer_TDTP · 2025-06-08
> >
> > Thank you for the author's response, which has basically addressed my main concern. I will increase my score.

---

### Official Review · Reviewer_J2Hz · 2025-05-12

**Rating:** 6
**Confidence:** 4
**Ethics Flag:** 1

**Summary:**

This paper tackles the hardware inefficiency of attention mechanisms during LLM decoding, specifically the bottleneck caused by large KV cache transfers and low arithmetic intensity. The authors propose two variants: Group-Tied Attention (GTA), which ties key and value states within groups to reduce KV cache size and improve arithmetic intensity over GQA; and Group Latent Attention (GLA), a parallel-friendly modification of latent attention (MLA) that shards the latent state to enable efficient tensor parallelism. The paper provides theoretical motivation based on arithmetic intensity, details system-level optimizations like asynchronous loading and distributed offset calculation for paged KV, and presents empirical results showing comparable/better model quality (perplexity, downstream tasks) for GTA/GLA against GQA/MLA respectively, along with significant kernel speedups (up to 2x for GLA vs. FlashMLA).

**Questions To Authors:**

- Could you elaborate on the practical implementation complexity of sharding the latent state and projections in GLA, especially for g > 2 shards?
- How does the communication overhead (e.g., AllReduce) in GLA with TP scale with the number of shards (g) and model size in practice, and how does it impact the observed kernel speedups in an end-to-end scenario?
- Table 1 shows arithmetic intensity. What specific operations and data types (e.g., BF16) are assumed in these calculations?

**Reasons To Accept:**

- Addresses a highly relevant and critical problem in LLM inference efficiency.
- Provides clear motivation grounded in hardware characteristics (arithmetic intensity).
- Proposes novel attention variants (GTA, GLA) with specific design goals (efficiency, parallelizability).
- Includes non-trivial system-level optimizations (distributed offset calculation) demonstrating practical implementation awareness.
- Strong empirical validation across multiple model scales, datasets, and metrics (quality, speed, parallelization scenarios) supporting the claims.
- GLA's design directly addresses a known limitation (parallelizability) of the recent MLA approach.

**Reasons To Reject:**

- The conceptual novelty of GTA could be seen as incremental over GQA (tying K/V).
- The speedup results focus on the attention kernel; end-to-end inference improvements are not directly reported (though the kernel is the bottleneck).
- Communication overhead associated with GLA's tensor parallelism (TP) is benchmarked separately but not deeply analyzed in the main evaluation sections.

---

> ### Author Response · Authors · 2025-06-03
> **Official Comment by Authors**
>
> We thank reviewer **J2Hz** for the thoughtful feedback on our work: "provides clear motivation grounded in hardware characteristics," proposes "novel attention variants (GTA, GLA)," includes "non-trivial system-level optimizations (distributed offset calculation) demonstrating practical implementation awareness," and offers "strong empirical validation across multiple model scales, datasets, and metrics (quality, speed, parallelization scenarios) supporting the claims." We address their remaining concerns below:
> >novelty of GTA incremental over GQA
>
> Our main contribution is the design principle of hardware-efficient attention: high arithmetic intensity and easy parallelism. Out of these principles, GTA is a result of a simple modification of GQA: tying K/V and applying decoupled RoPE. We would argue that this simplicity is a strength. MQA was also a simple change over MHA: just tying all KV heads together. GQA was another simple change: instead of tying all the heads, only tie heads in the same group. Thanks to this simplicity, GQA has become the standard attention variant in open-source models, predominantly due to its grouping nature, which leads to a parallel-friendly design.
>
> GTA employs a similar grouping style for ease of parallelization but, to our knowledge, is the first to tie KV to a single state and preserve quality. It exploits the very low-rank subspace of pre-RoPE keys while maintaining a separate single-head half-dimension RoPE to preserve positional information and, therefore, quality. Through extensive ablation, we found that RoPE should remain free from the shared half-component of K and V when the rotation is applied to the keys and immediately inverted before reusing it for the value path. The tied portion is never rotated for the keys to avoid performance degradation, which could potentially interfere with semantic content. GTA yields notable savings: on Llama 3 8B with TP 4, GTA 8 cuts KV cache by 37.5% versus GQA 8, enabling longer contexts and larger batches. In a 1.471 billion-parameter experiment, GTA 4 halves the KV cache at TP 2 while matching GQA 4 on perplexity (10.129 vs 10.202) and downstream accuracy (60.2%).
>
> > E2E inference and TP communication overhead
>
> This concern is valid. We conducted comprehensive benchmarks measuring E2E latency and throughput in a live server, capturing All Reduce communication and down to HTTP overhead, reflecting real deployment conditions. Full results appear in the general response and are detailed in the anonymous link below; we briefly summarize the results here.
>
> * 64-request throughput GLA-8 TP8, with latent dimension $d_c$=256 for 8 heads sharded across 8 GPUs achieves 2x the throughput of MLA TP8 with $d_c=512$. GLA-8 TP8 outperforms MLA TP2 DP4 as well as under an equivalent hybrid TP + DP setup.
> * Workload imbalance (prefill ≤ 131 K & decode ≤ 4K) where the prefill lengths are sampled uniformly up to 131K tokens, GLA-8 achieves roughly 2.5× the MLA throughput (TP= 2, DP=4). GLA-4 (TP=4, DP=2) also exceeds MLA under (TP=2, DP=4) and remains more tolerant of workload imbalance due to its lower DP rank
> * With 16 requests, prefilling 32 K or 64 K, and decoding 4 K, with each prefill length fixed to 32 K or 64 K tokens and the decoding length fixed to 4 K, GLA-8 with pure TP=8 achieves higher throughput than MLA under hybrid parallelism.
> * Overall scalability Across every tested pure-TP or low-DP configuration, zero-redundancy sharded GLA or when GLA's latent heads are smaller than the TP rank (some duplication occurs but less than MLA) matches or surpasses MLA, boosting peak throughput and resisting straggler effects.
>
> > practical implementation complexity of sharding the latent for g > 2 shards?
>
> GLA compressed tokens to $h_c$ latent heads of dimension $d_c = 2d_h$, half of MLA's $4d_h$. During training each latent head and its associated up projection matrices reconstruct distinct KV features for its $g_q = h_q/h_c$ query heads, so each up projection has $g_q\ d_h$ columns instead of MLA's $h_q\ d_h$. TP partitions the key and value up-projections in column-parallel fashion across ranks, $W^{UK}, W^{UV}\in\mathbb{R}^{2d_h\times g_q d_h}$. After weight absorption at decoding, each latent head attends only to its group. Distributing latent heads across $TP$ ranks yields head level parallelism while duplicating the latent KV cache at most $TP/h_c$ when $h_c \le \text{TP}$; duplication is zero when $h_c = TP$.
>
> >Table 1 arithmetic intensity operations and data types (e.g., BF16)?
>
> The arithmetic intensity in Table 1 assumes BF16 elements and counts the FLOPs in (Q Kᵀ)V during decoding. Shifting to FP8 halves the bytes per element and doubles the peak tensor-core throughput on most accelerators, so both the kernel intensity and the machine crossover double. Their ratio is preserved, meaning the optimal grouping and tying choices remain unchanged; they run faster.

---

> > ### Comment · Reviewer_J2Hz · 2025-06-09
> >
> > Thanks for the response, the anwser addressed my questions, I would to raise the score. Please include these contents into the paper.

---

### Official Review · Reviewer_CoaW · 2025-05-21

**Rating:** 7
**Confidence:** 4
**Ethics Flag:** 1

**Summary:**

This paper introduces two hardware-efficient attention mechanisms, Group Tied Attention (GTA) and Group Latent Attention (GLA), designed to accelerate Large Language Model (LLM) decoding by addressing bottlenecks related to data movement, expanding Key-Value (KV) caches, and limited parallelism.

**Questions To Authors:**

1. How does GTA/GLA affects retrieval intensive tasks performance? What about long contecxt?

**Reasons To Accept:**

1. The paper tackles significant and well-recognized challenges in LLM inference: excessive data movement, the expanding KV cache, and limited parallelism in decoding.
2. The authors ground their designs in an analysis of arithmetic intensity, providing a clear rationale for how their methods improve hardware efficiency.
3. The authors details crucial low-level optimizations like asynchrony through software pipelining/warp specialization and a novel distributed offset calculation for paged KV. The latter significantly improves paged KV performance, especially for small page sizes.
4. The experiments cover multiple model scales, datasets, and evaluation metrics (perplexity, downstream tasks, latency, hardware utilization), providing a robust assessment of the proposed methods.

**Reasons To Reject:**

1. The largest scale of the experiment is around 900M parameters with 50B tokens, which is relatively small. The downstream task performances comparing MLA to GLA is inconsistant across scales (i.e. GLA is better at 433M but worse in 876M ). I would suggest having a few 1.3B/100B setting experiments to see a clearer scaling trend.

---

> ### Author Response · Authors · 2025-06-03
> **Official Comment by Authors**
>
> We thank **CoaW**  for their positive review of our work, noting that our papers “experiments cover multiple model scales, datasets, and evaluation metrics (perplexity, downstream tasks, latency, hardware utilization), providing a robust assessment of the proposed methods.” Also, we appreciate them noting that “the authors ground their designs in an analysis of arithmetic intensity, providing a clear rationale for how their methods improve hardware efficiency”
>
> >The largest scale of the experiment is around 900M parameters with 50B tokens, which is relatively small. The downstream task performances comparing MLA to GLA is inconsistent across scales (i.e. GLA is better at 433M but worse in 876M ). I would suggest having a few 1.3B/100B setting experiments to see a clearer scaling trend.
>
> During rebuttal, we trained a 1.5B-parameter (next size in GPT-3 config) on 50B tokens to test scalability with Llama-3 architecture. Each variant (GQA, GTA, MHA, MLA, and GLA) ran about 2.5 days on eight H100s; doubling to 100B tokens would cost roughly        $ \textdollar 14,000 $ at $ \textdollar 3 $ per H100-hour across the five attention variants, exceeding budget and time (taking twice as long). We report FineWeb-Edu validation perplexity, average perplexity across five datasets, downstream accuracy on seven tasks, and per-token KV cache for TP=1 and 2. GLA-2 (two latent heads) attains perplexity 10.218 and 60.0% accuracy vs. MLA’s 10.256 and 59.1%, while halving per-device KV cache at TP=2. GTA-4 records 10.129 and 60.2% vs. GQA-4’s 10.202 and 60.2%, also halving KV cache at TP=2.
>
> |Method|Fine-Web PPL|Avg. PPL |Avg. DS| KV cache TP1|KV cache TP2|
> |---|---|---|---|---|---|
> |MHA|10.311|21.206|60.1|8192|4096|
> |GQA-4|10.202|21.073|60.2|2048|1024|
> |GTA-4|10.129|20.823|60.2|1152|640|
> |GLA-2|10.218|21.163|60|1152|640|
> |MLA|10.256|21.199|59.1|1152|1152|
>
> >How does GTA/GLA affect retrieval intensive tasks performance? What about long context?
>
> GTA and GLA are similar in spirit to GQA and MLA, unlike attention approximations such as sliding window or linear attention that differ more significantly from MHA. As a result, we do not expect much difference in quality compared to GQA and MLA. Our evaluations have validated this for ~7 general tasks, and we plan to train longer context models to validate the long context ability.

---

> > ### Comment · Reviewer_CoaW · 2025-06-09
> >
> > Thanks for the added experiments. I will raise my score to 7.

---

### Official Review · Reviewer_R9T9 · 2025-05-23

**Rating:** 8
**Confidence:** 5
**Ethics Flag:** 1

**Summary:**

This paper propose novel attention variant GTA(Group Tied Attention) and GLA(Group Latent Attention) for decoding efficiency. Multi-Head Attention (MHA) is known for low arithmetic intensity during decode stage, GQA (Grouped Query Attention) and MLA (Multi-Latent Attention) increase arithmetic intensity. However, MLA do not support tensor parallelism when used in distributed inference because the number of key/value head is only one and can not be further sharded on head dimension. The GLA proposed in this paper addresses this issue by combining GQA and MLA, splitting head dimension into groups and breaks the data dependency of different groups to make GLA compatible with TP. GLA designs efficient CUDA kernel pipeline for Hopper and achieve significant speedup in compute-bound setting (MTP enabled). Evaluations on downstream tasks show competitive performance compared to alternatives.

**Questions To Authors:**

* Can we adapt existing models to GLA without training or with post-training, like in TransMLA(https://arxiv.org/abs/2502.07864)
* How does GLA compare to GQA with higher group size (such as 32)?
* Where does kernel performance over MLA coming from? I can imagine that compared to MLA, GLA don't need to broadcast QK results to all warps participated in PV computation, thus requiring less synchronization, but not sure if it will be the same case for Blackwell, where   mma output already stored in TMEM.
* page_size = 1 and MLA pipeline design is also explored in flashinfer, which is also worth mentioning in related work.

**Reasons To Accept:**

* A simple and effective extension to MLA, address an important issue that MLA is not compatible with TP.
* Train Llama-3 architecture (maximum scale at 876M) with GLA from scratch on 100B scale datasets, experiments on down-stream tasks show competitive performance.
* Kernel optimization for GLA, higher TFLOPs/s than MLA. Distributed offset calculation to alleviate pointer arithmetic issue for LDGSTS style data-loading.

**Reasons To Reject:**

* The advantage over GQA is not significant, lack study of comparing GLA with GQA with higher group ratio.

---

> ### Author Response · Authors · 2025-06-03
> **Official Comment by Authors**
>
> We would like to thank **R9T9** for their positive feedback and for highlighting that “effective extension to MLA, addresses an important issue that MLA is not compatible with TP”. We also express gratitude towards them for emphasizing positively that our “Kernel optimization for GLA, higher TFLOPs/s than MLA. Distributed offset calculation to alleviate pointer arithmetic issue for LDGSTS style data-loading.”
>
> >The advantage over GQA is not significant, lack study of comparing GLA with GQA with higher group ratio. How does GLA compare to GQA with higher group size (such as 32)?
>
> For the 1.5B model (the next GPT-3 configuration step), GLA-2 (with two latent heads) achieves a validation perplexity of 10.21 and an average downstream accuracy of 60.0% across seven tasks, while GQA-4 yields 10.20 and 60.2%. Without sharding, their KV caches consume 1052 and 2048 bytes per token; at TP = 2 the figures fall to 640 and 1024. GLA-2 matches GQA-4 at half the logical KV cache.
>
> We benchmark three GQA variants, GQA-1 (MQA), GQA-4, and GQA-16 (MHA), on most of our models. Model sizes ranging from 433M-1.5B, all have 16 query heads. GLA-2 matches or tops every variant, so ablations are not informative until we reach larger scales, such as GPT-3 with a 2.7B model scale with 32 query heads.
>
> Looking ahead, the Llama 4 family (up to 400 B) employs GQA 8, whose KV cache splits cleanly across eight GPUs ($2\ d_h$ bytes per token per device at TP=8). A matching GLA-8 would store about $2.5\ d_h$ because a decoupled RoPE adds $\frac{d_h}{2}$. This extra overhead can be reduced by applying RoPE only to some layers, as current Llama models already do. Whether GLA-8 improves quality over GQA-8 with comparable memory budgets remains an open question. However, at the moment, at a 1.5B scale, it matches GQA with half the bytes allocated to caching previously generated tokens.
>
> |Model Size|#Param|#Layer|d_model|$h_q$|$d_h$|
> |---|---|---|---|---|---|
> |Small|183.65M|12|768|12|64|
> |Medium|433.77M|24|1024|16|64|
> |Large|876.55M|24|1536|16|96|
> |XL|1471.12M|24|2048|16|128|
>
> >Can we adapt existing models to GLA without training or with post-training, like in TransMLA?
>
> Yes, GQA can be distilled into GLA with post-training using similar approaches mentioned in TransMLA. GLA allows the up-projections to be dense, so it can mix latent dimensions in ways that plain GQA cannot. When those extra degrees of freedom are used, the mapping no longer goes in the opposite direction.
>
> >Where does kernel performance over MLA coming from? I can imagine that compared to MLA, GLA don't need to broadcast QK results to all warps participated in PV computation, thus requiring less synchronization, but not sure if it will be the same case for Blackwell, where mma output already stored in TMEM.
>
> The reasons the GLA kernel is faster than the MLA kernel are:
> (1) for speculative decoding / MTP setup with seqlen_q >= 2, MLA becomes much more compute bound due to the large head dimension (512/576), while GLA hits the sweet spot of maxing out compute & memory bandwidth with head dimension 256 / 320 (and double the number of heads). We expect this trend to hold for Blackwell since the ratio of max FLOPS / max mem bandwidth is around the same (989TFLOPS / 3.35TB/s = 295 for H100 and 2500TFLOPS / 8TB/s = 312 for GB200).
> (2) For the non-speculative decoding setup (seqlen_q = 1), as the reviewer has observed, GLA does not need to broadcast QK results to all warps. For Blackwell, TMEM replaces the role of registers to store mma output, so this might be less of a difference. However, TMEM size is still limited by 256KB (128 rows, at most 512 columns, each storing 4 bytes). For MLA with output head dimension of 512, if we use tile_M = 128, just storing the output accumulator would already max out the TMEM, leaving no space for the accumulator of the QK mma. As a result, one would need to use tile_M = 64 for MLA. For GLA with head dimension 256 / 320, one could use tile_M = 128. As we are excited about this direction, we are currently writing Blackwell kernels to explore how much of a difference this would make.
>
> >page_size = 1 and MLA pipeline design is also explored in flashinfer, which is also worth mentioning in related work.
>
>
> We appreciate **R9T9** for suggesting this, we will include FlashInfer citation in our camera ready version.

---

### Author Response · Authors · 2025-06-03
**Official Comment by Authors**

We appreciate the thoughtful and positive feedback from the reviewers. We are encouraged by the reviewers for praising our timely LLM inference study for grounding in arithmetic intensity & hardware characteristics [TDTP, CoaW, J2Hz], its non-trivial system optimizations, including a novel distributed offset for paged KV [R9T9], and its strong empirical validation across multiple model scales with preserved generation quality [CoaW, J2Hz, TDTP]. Since the submission, encouraged by our positive results, we have scaled up our train-from-scratch experiments and performed extensive end-to-end inference benchmarks. The same trend reported in the paper’s evaluation section continues to hold, and we included these new results below.

The core objective of our work is to rethink attention through the lens of arithmetic intensity to reduce the KV cache size, thereby accelerating decoding without compromising model quality or scalability in distributed inference. Our contributions are: 1) Grouped Latent Attention & Grouped-Tied Attention reorganize QKV interaction via grouping & tying to utilize hardware effectively; 2) Systems-level optimization (sync pipelining, warp specialization, and a distributed offset calculator) turn the design into deployable code; and 3) Extensive experiments (new E2E benchmarks & 1.5B scale experiments extend evidence).

The primary concern among reviewers is insufficient evidence for GLA’s E2E performance and quality at a larger scale. J2HZ noted E2E gains are unclear, e.g., considering TP communication overhead, while TDTP mentioned acceleration is for specific scenarios, and CoaW suggested 1.3B model experiments for further evidence. We added live-server benchmarks and larger-scale tests to address these points. We have added an 11-page appendix that now reports GLA speed on live-server production benchmarks, which include TP communication, dynamic queueing, network, and HTTP parsing overheads, reflecting real-world deployment. We are excited to report that: (1) GLA delivers up to 2x the throughput and half the latency by fetching a smaller KV cache per device, achieved by grouping and sharding the latent head. (2) Also, quality scales to 1.5B: GTA outperforms GQA with half the KV cache, and GLA tops MLA.

For the end-to-end (E2E) benchmarks, all experiments run in SGLang live-server mode on x8 H100 GPUs, using DeepSeek Coder V2 Base (236B→21B active), FP8, page 64, and 8192 chunked prefill. A load generator issues 1280 prompts with a maximum concurrency limiting the number of active requests. We compare pure TP with a hybrid TP+DP scheme that applies DP only to attention to mitigate MLA KV duplication. Results are reported in the tables below for GLA vs. MLA: median E2E latency, TTFT, ITL, and throughput across six workload patterns
Also visualization and additional benchmark (e.g decode heavy) results appear in the following link: https://purring-wealth-22c.notion.site/COLM-Rebuttal-204dcbebba6e80af8dd6c77a9b149aaf?source=copy_link

# E2E live server benchmark
* ## TP  &  TP+DP (Prefill/Decode 8K/4K)
**64 max-conc.**

| Method | E2E Latency (s) | TTFT (s) | ITL (ms) | Throughput (tok/s) |
|-|-|-|-|-|
| GLA-8 (TP8) | 179.32 | 11.96 | 38.16 | 1460.61 |
| MLA (TP8) | 381.13 | 192.70 | 43.03 | 858.95 |
| GLA-2 (TP2 DP4) | 165.86 | 14.12 | 35.01 | 1583.51 |
| MLA (TP2 DP4) | 196.47 | 14.78 | 42.35 | 1334.18 |
| GLA-4 (TP4 DP2) | 170.66 | 12.80 | 36.07 | 1542.96 |
| MLA (TP4 DP2) | 205.39 | 13.36 | 44.51 | 1276.25 |

* ## Latency-sensitive (Prefill/Decode 64K/256 with max-conc. 3)

| Method | E2E Latency (s) | TTFT (s) | ITL (ms) |Throughput (tok/s) |
|-|-|-|-|-|
| GLA-8 (TP8) | 24.60 | 12.96 | 24.54 | 31.17 |
| MLA (TP8 DP4) | 54.25 | 46.76 | 28.14 | 14.14 |

* ## Workload Imbalance (Var. seq. length max-conc. 4)

*prefill ~U[16K,131K] & decode ~U[512, 4K]*

| Method | E2E Latency (s) | TTFT (s) | ITL (ms) | Throughput (tok/s) |
|-|-|-|-|-|
| GLA-8 (TP8) | 89.57 | 7.58 | 25.45 | 100.68 |
| MLA (TP2 DP4) | 233.69 | 38.54 | 28.66 | 37.20 |

*prefill ~U[4K, 32K] & decode ~U[512, 4K]*

| Method | E2E Latency (s) | TTFT (s)|  ITL (ms) | Throughput (tok/s) |
|-|-|-|-|-|
| GLA-8 (TP8)  | 55.97 | 1.14 | 22.32 | 165.78 |
| MLA (TP2 DP4)  | 73.82 | 3.29 | 25.73 | 125.31 |

* ## Short chat
**1 max-conc. w/ Prefill/Decode 256/128**

| Method |E2E Latency (s)| TTFT (s)| ITL (ms)| Throughput (tok/s) |
|-|-|-|-|-|
| GLA-8 (TP8) | 2.49 | 0.11 | 18.72 | 51.45 |
| MLA (TP2 DP4)| 2.91 | 0.12  | 21.94 | 43.96 |

**8 max-conc. w/ Prefill/Decode 2k/2k**
| Method | E2E Latency (s) | TTFT (s) | ITL (ms) | Throughput (tok/s) |
|-|-|-|-|-|
| GLA-8 (TP8) | 47.18  | 0.86 | 22.54  | 346.92|
| MLA (TP2 DP4) | 56.35 | 0.82 | 27.04  | 290.91|

# 1.5B for 50Bt
|Method|Fine-Web PPL|Avg. PPL |Avg. DS| KV cache TP1|KV cache TP2|
|-|-|-|-|-|-|
|MHA|10.311|21.206|60.1|8192|4096|
|GQA-4|10.202|21.073|60.2|2048|1024|
|GTA-4|10.129|20.823|60.2|1152|640|
|GLA-2|10.218|21.163|60|1152|640|
|MLA|10.256|21.199|59.1|1152|1152|

---

### Decision · Program_Chairs · 2025-07-08

**Decision:**

Accept

**Comment:**

This paper presents a timely and valuable contribution to the field of efficient Large Language Model inference. It introduces two novel attention variants, Group-Tied Attention (GTA) and Group Latent Attention (GLA), designed to alleviate critical bottlenecks in LLM decoding, namely excessive data movement, expanding KV caches, and limited parallelism. The work is commendably grounded in a hardware-aware perspective, focusing on optimizing arithmetic intensity and enabling effective parallelization. The authors motivate their architectural modifications through a clear analysis of arithmetic intensity and hardware characteristics. This provides a solid foundation for understanding why these variants are expected to be more efficient. GTA ties key and value states to reduce KV cache and improve arithmetic intensity over Grouped Query Attention, while GLA modifies Multi-head Latent Attention to be more parallel-friendly by sharding the latent state, a key limitation of prior latent attention methods.